# Learning Equilibria in Matching Markets from Bandit Feedback

**Meena Jagadeesan**[†]
UC Berkeley
mjagadeesan@berkeley.edu

**Alexander Wei**[†]
UC Berkeley
awei@berkeley.edu

**Yixin Wang**
UC Berkeley
ywang@eecs.berkeley.edu

**Michael I. Jordan**
UC Berkeley
jordan@cs.berkeley.edu

**Jacob Steinhardt**
UC Berkeley
jsteinhardt@berkeley.edu

## Abstract

Large-scale, two-sided matching platforms must find market outcomes that align with user preferences while simultaneously learning these preferences from data. But since preferences are inherently uncertain during learning, the classical notion of stability (Gale and Shapley, 1962; Shapley and Shubik, 1971) is unattainable in these settings. To bridge this gap, we develop a framework and algorithms for learning stable market outcomes under uncertainty. Our primary setting is matching with transferable utilities, where the platform both matches agents and sets monetary transfers between them. We design an incentive-aware learning objective that captures the distance of a market outcome from equilibrium. Using this objective, we analyze the complexity of learning as a function of preference structure, casting learning as a stochastic multi-armed bandit problem. Algorithmically, we show that "optimism in the face of uncertainty," the principle underlying many bandit algorithms, applies to a primal-dual formulation of matching with transfers and leads to near-optimal regret bounds. Our work takes a first step toward elucidating when and how stable matchings arise in large, data-driven marketplaces.[1]

## 1 Introduction

Data-driven marketplaces face the simultaneous challenges of learning agent preferences and aligning market outcomes with the incentives induced by these preferences. Consider, for instance, online platforms that match two sides of a market to each other (e.g., Lyft, TaskRabbit, and Airbnb). On these platforms, customers are matched to service providers and pay for the service they receive. If agents on either side are not offered desirable matches at fair prices, they would have an incentive to leave the platform and switch to a competing platform. Agent preferences, however, are often unknown to the platform and must be learned. When faced with uncertainty about agent preferences (and thus incentives), *when can a marketplace efficiently explore and learn market outcomes that align with agent incentives?*

We center our investigation around a model called *matching with transferable utilities*, proposed by Shapley and Shubik [32]. In this model, there is a two-sided market of customers and service providers. Each customer has a utility that they derive from being matched to a given provider and vice versa. The platform selects a matching between the two sides and assigns a monetary transfer between each pair of matched agents. Transfers are a salient feature of most real-world matching

---

[†]Equal contribution.

[1]A full version of this paper (referenced as [18]) is available at `https://arxiv.org/abs/2108.08843`.

markets: riders pay drivers on Lyft, clients pay freelancers on TaskRabbit, and guests pay hosts on Airbnb. An agent's net utility is their value for being matched to their partner plus the value of their transfer (either of which can be negative in the cases of costs and payments). In matching markets, the notion of *stability* captures alignment of a market outcome with agent incentives. Informally, a market outcome is *stable* if no pair of agents would rather match with each other than abide by the market outcome, and stable matchings can be computed when preferences are fully known.

However, in the context of large-scale matching platforms, the assumption that preferences are known breaks down. Platforms usually cannot have users report their complete preference profiles. Moreover, users may not even be aware of what their own preferences are. For example, a freelancer may not exactly know what types of projects they prefer until actually trying out specific ones. In reality, a data-driven platform is more likely to learn information about preferences from repeated feedback over time. Two questions now emerge: In such marketplaces, how can stable matchings be learned? And what underlying structural assumptions are necessary for efficient learning to be possible?

Toward answering these questions, we propose and investigate a model for learning stable matchings from noisy feedback. We model the platform's learning problem using stochastic multi-armed bandits, which lets us leverage the extensive body of techniques in the bandit literature to analyze the data efficiency of learning. Specifically, our main contributions are: (i) We develop an incentive-aware learning objective—Subset Instability—that captures the distance of a market outcome from equilibrium. (ii) Using Subset Instability as a measure of regret, we show that any "UCB-based" algorithm can be adapted to this incentive-aware setting. (iii) We instantiate this idea for two families of preference structures to design efficient algorithms for incentive-aware learning, helping elucidate how preference structure affects the complexity of learning stable matchings.

**Designing the learning objective.** Since mistakes are inevitable while exploring and learning, achieving exact stability at every time step is an unattainable goal. To address this issue, we lean on approximation, focusing on learning market outcomes that are *approximately* stable. Thus, we need a metric that captures the distance of a market outcome from equilibrium.

We introduce a notion for approximate stability that we call Subset Instability. Specifically, we define the Subset Instability of a market outcome to be the maximum difference, over all subsets $\mathcal{S}$ of agents, between the total utility of the maximum weight matching on $\mathcal{S}$ and the total utility of $\mathcal{S}$ under the market outcome. We show Subset Instability can be interpreted as how much the platform would have to *subsidize* participants to keep them on the platform and make the resulting matching stable. We also show that Subset Instability is the maximum gain in utility that a coalition of agents could have derived from an alternate matching such that no agent in the coalition is worse off.

Subset Instability also satisfies the following properties, which make it suitable for learning: (i) Subset Instability is equal to $0$ if and only if the market outcome is (exactly) stable; (ii) Subset Instability is robust to small perturbations to the utility functions of individual agents, which is essential for learning with noisy feedback; (iii) Subset Instability upper bounds the utility difference of a market outcome from the socially optimal market outcome.

**Designing algorithms for learning a stable matching.** Using Subset Instability, we investigate the problem of learning a stable market outcome from noisy user feedback in a stochastic bandit model. In each round, the platform selects a market outcome (i.e., a matching along with transfers), with the goal of minimizing cumulative instability.

We develop a general approach for designing bandit algorithms within our framework. Our approach is based on a primal-dual formulation of matching with transfers [32] We find that "optimism in the face of uncertainty," the principle underlying many UCB-style bandit algorithms [3, 22], can be adapted to this primal-dual setting. The resulting algorithm is simple: maintain upper confidence bounds on the agent utilities and compute, in each round, an optimal primal-dual pair in terms of these upper confidence bounds. Following this approach, we show regret bounds for two sets of structural assumptions on agent preferences, which we now state.

**Theorem 1.1** (Unstructured Preferences, Informal). *There exists a UCB-style algorithm that incurs $\widetilde{O}(N\sqrt{nT})$ regret according to Subset Instability after $T$ rounds, where $N$ is the number of agents on the platform and $n$ is the number of agents that arrive in any round. (In fact, this bound is optimal up to logarithmic factors.)*

**Theorem 1.2** (Separable Linear Preferences, Informal). *Suppose each agent $a$ gets utility $\langle \phi(a), c_{a'} \rangle$ from matching with $a'$, where $\phi(a) \in \mathbb{R}^d$ is unknown and $c_{a'} \in \mathbb{R}^d$ is known. Then there exists a UCB-style algorithm that incurs $O(d\sqrt{N}\sqrt{nT})$ regret according to Subset Instability after $T$ rounds, for $N$ the number of agents on the platform and $n$ the maximum number of agents in any round.*

These results highlight the role of preference structure on the complexity of learning a stable matching. To see this concretely, consider the case where all agents show up each round, so $n = N$. In this case, our regret bound for unstructured preferences is superlinear in $N$; and this dependence on $N$ is *necessary* due to a lower bound (see Lemma 5.4). On the other hand, our regret bound is linear in $N$ for separable linear preferences. This means that in large markets, a centralized platform can efficiently learn a stable matching with this preference structure assumption.

**Extensions.** In the full version [18], we provide several extensions of our framework and results to additional classes of preference structures, alternative forms of regret bounds, and richer platform objectives, and other models of matching markets. In Section 6, we briefly discuss two of these extensions—instance-dependent regret bounds and matching markets with non-transferable utilities.

## 1.1 Related work

In the machine learning literature, starting with Das and Kamenica [11] and Liu et al. [24], several works [11, 24, 30, 25, 7, 6] study learning stable matchings from bandit feedback in the Gale-Shapley stable marriage model [15]. A major difference between this setting and ours is the absence of monetary transfers between agents. These works focus on the *utility difference* rather than the instability measure that we consider. Cen and Shah [7] extend this bandits model to incorporate fixed, predetermined cost/transfer rules. However, they do not allow the platform to set arbitrary transfers between agents. Moreover, they also consider a weaker notion of stability that does not consider agents negotiating arbitrary transfers: defecting agents must set their transfers according to a fixed, predetermined structure. In contrast, we follow the classical definition of stability [32].

Outside of machine learning, several papers also consider the complexity of finding stable matchings in other feedback and cost models, e.g., communication complexity [16, 2, 33] and query complexity [12, 2]. Of these works, Shi [33], which studies the communication complexity of finding approximately stable matchings with transferable utilities, is most similar to ours. Shi assumes agents know their preferences and focuses on the communication bottleneck, whereas we study the costs associated with learning preferences. Moreover, the approximate stability notion in Shi [33] is the maximum unhappiness of any *pair* of agents, whereas Subset Instability measures the maximum unhappiness over any *subset* of agents. For learning stable matchings, Subset Instability has the advantages of being more fine-grained and having a primal view that motivates a clean UCB-based algorithm.

Our notion of instability connects to historical works in coalitional game theory: related are the concepts of the strong-$\varepsilon$ core of Shapley and Shubik [31] and the indirect function of Martínez-Legaz [27], although each was introduced in a very different context than ours. Nonetheless, they reinforce the fact that our instability notion is a very natural one to consider.

Multi-armed bandits have also been applied to learning in other economic contexts. For example, learning a socially optimal matching (without learning transfers) is a standard application of combinatorial bandits [8, 14, 9, 10, 21]. Other applications at the interface of bandit methodology and economics include dynamic pricing [28, 20, 5], incentivizing exploration [13, 26], learning under competition [1], and learning in matching markets without incentives [19].

Finally, primal-dual methods have also featured elsewhere in the bandits literature (e.g., [17, 34, 23]).

## 2 Preliminaries

The foundation of our framework is the *matching with transfers* model of Shapley and Shubik [32]. In this section, we introduce this model along with the concept of stable matching.

### 2.1 Matching with transferable utilities

Consider a two-sided market that consists of a finite set $\mathcal{I}$ of customers on one side and a finite set $\mathcal{J}$ of providers on the other. Let $\mathcal{A} := \mathcal{I} \cup \mathcal{J}$ be the set of all agents. A *matching $X \subseteq \mathcal{I} \times \mathcal{J}$ is*

a set of pairs $(i, j)$ that are pairwise disjoint, representing the pairs of agents that are matched. Let $\mathscr{X}_\mathcal{A}$ denote the set of all matchings on $\mathcal{A}$. For notational convenience, we define for each matching $X \in \mathscr{X}_\mathcal{A}$ an equivalent functional representation $\mu_X : \mathcal{A} \to \mathcal{A}$, where $\mu_X(i) = j$ and $\mu_X(j) = i$ for all matched pairs $(i, j) \in X$, and $\mu_X(a) = a$ if $a \in \mathcal{A}$ is unmatched.

When a pair of agents $(i, j) \in \mathcal{I} \times \mathcal{J}$ matches, each experiences a utility gain. We denote these utilities by a global utility function $u : \mathcal{A} \times \mathcal{A} \to \mathbb{R}$, where $u(a, a')$ denotes the utility that agent $a$ gains from being matched to agent $a'$. (If $a$ and $a'$ are on the same side of the market, we take $u(a, a')$ to be 0 by default.) We allow these utilities to be negative, if if matching results in a net cost (e.g., if an agent is providing a service). We assume each agent $a \in \mathcal{A}$ receives zero utility if unmatched, i.e., $u(a, a) = 0$. When we wish to emphasize the role of an individual agent's utility function, we will use the equivalent notation $u_a(a') \coloneqq u(a, a')$.

A *market outcome* consists of a matching $X \in \mathscr{X}_\mathcal{A}$ along with a vector $\tau \in \mathbb{R}^\mathcal{A}$ of transfers, where $\tau_a$ is the amount of money transferred from the platform to agent $a$ for each $a \in \mathcal{A}$. These monetary transfers are a salient feature of most real-world matching markets: riders pay drivers on Lyft, clients pay freelancers on TaskRabbit, and guests pay hosts on Airbnb. Shapley and Shubik [32] capture this aspect of matching markets by augmenting the classical two-sided matching model with transfers of utility between agents. Transfers are typically required to be *zero-sum*, meaning that $\tau_i + \tau_j = 0$ for all matched pairs $(i, j) \in X$ and $\tau_a = 0$ if $a$ is unmatched. Here, $X$ represents how agents are matched and $\tau_a$ represents the transfer that agent $a$ receives (or pays). The net utility that an agent $a$ derives from a matching with transfers $(X, \tau)$ is therefore $u(a, \mu_X(a)) + \tau_a$.

**Stable matchings.** In matching theory, stability captures when a market outcome aligns with individual agents' preferences. Roughly speaking, a market outcome $(X, \tau)$ is stable if: (i) no individual agent $a$ would rather be unmatched, and (ii) no pair of agents $(i, j)$ can agree on a transfer such that both would rather match with each other than abide by $(X, \tau)$. Formally:

**Definition 2.1.** A market outcome $(X, \tau)$ is *stable* if: (i) it is *individually rational*, i.e.,

$$u_a(\mu_X(a)) + \tau_a \geq 0 \tag{1}$$

for all agents $a \in \mathcal{A}$, and (ii) it *has no blocking pairs*, i.e.,

$$\big(u_i(\mu_X(i)) + \tau_i\big) + \big(u_j(\mu_X(j)) + \tau_j\big) \geq u_i(j) + u_j(i) \tag{2}$$

for all pairs of agents $(i, j) \in \mathcal{I} \times \mathcal{J}$.[2]

A fundamental property of the matching with transfers model is that if $(X, \tau)$ is stable, then $X$ is a maximum weight matching, i.e., $X$ maximizes $\sum_{a \in \mathcal{A}} u_a(\mu_X(a))$ over all matchings $X \in \mathscr{X}_\mathcal{A}$ [32]. The same work shows that stable market outcomes coincide with Walrasian equilibria.

To make the matching with transfers model concrete, we use the simple market depicted in the center panel of Figure 1 as a running example throughout the paper. This market consists of a customer Charlene and two providers Percy and Quinn, which we denote by $\mathcal{I} = \{C\}$ and $\mathcal{J} = \{P, Q\}$. If the agents' utilities are as given in Figure 1, then Charlene would prefer Quinn, but Quinn's cost of providing the service is much higher. Thus, matching Charlene and Percy is necessary for a stable outcome. This matching is stable for any transfer from Charlene to Percy in the interval $[5, 7]$.

## 3 Learning Problem and Feedback Model

We instantiate the platform's learning problem in a stochastic contextual bandits framework. Matching takes place over the course of $T$ rounds. We denote the set of all customers by $\mathcal{I}^*$, the set of all providers by $\mathcal{J}^*$, and the set of all agents on the platform by $\mathcal{A}^* = \mathcal{I}^* \cup \mathcal{J}^*$. Each agent $a \in \mathcal{A}^*$ has an associated context $c_a \in \mathcal{C}$, where $\mathcal{C}$ is the set of all possible contexts. The context $c_a$ represents the side information available to the platform about $a$ (e.g., demographic or location information). Each round, a set of agents arrives to the market. The platform then selects a market outcome and incurs a regret equal to the *instability* of the market outcome (which we introduce formally in Section 4). Finally, the platform receives noisy feedback about the utilities of each matched pair.

---

[2]We observe that (2) corresponds to no pair of agents $(i, j)$ being able to agree on a transfer such that both would rather match with each other than abide by $(X, \tau)$. Notice that a pair $(i, j)$ violates (2) if and only if they can find a transfer $\tau_i' = -\tau_j'$ such that $u_i(j) + \tau_i' > u_i(\mu_X(i)) + \tau_i$ and $u_j(i) + \tau_j' > u_j(\mu_X(j)) + \tau_j$.

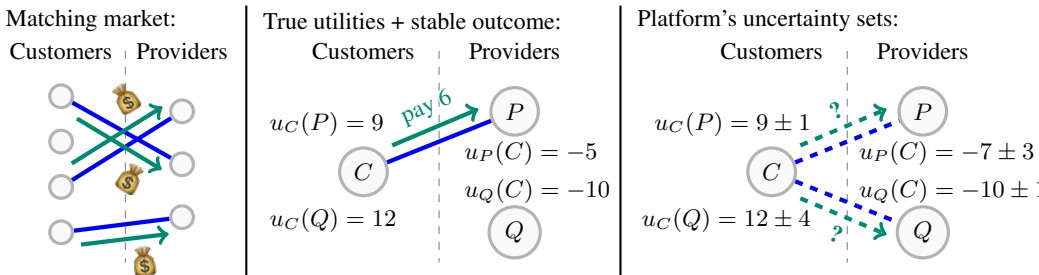

Figure 1: The left panel depicts a schematic of a matching (blue) with transfers (green). The center panel depicts a matching market with three agents and a stable matching with transfers for that market. (If the transfer 6 is replaced with any value between 5 and 7, the outcome remains stable.) The right panel depicts the same market, but with utilities replaced by uncertainty sets; note that no matching with transfers is stable for all realizations of utilities.

To interpret the noisy feedback, note that platforms in practice often receive feedback both explicitly (e.g., riders rating drivers after a Lyft ride) and implicitly (e.g., engagement metrics on an app). In either instance, feedback is likely to be sparse and noisy. For simplicity, we do not account for agents strategically manipulating their feedback to the platform and focus on the problem of learning preferences from unbiased reports.

We now describe this model more formally. In the $t$-th round:

1. A set $\mathcal{I}^t \subseteq \mathcal{I}^*$ of customers and a set $\mathcal{J}^t \subseteq \mathcal{J}^*$ of providers arrive to the market. Write $\mathcal{I}^t \cup \mathcal{J}^t =: \mathcal{A}^t$. The platform observes $\mathcal{A}^t$ and the *context* $c_a \in \mathcal{C}$ of each agent $a \in \mathcal{A}^t$.
2. The platform selects a matching with *zero-sum* transfers $(X^t, \tau^t)$ between $\mathcal{I}^t$ and $\mathcal{J}^t$.
3. The platform observes noisy utilities $u_a(\mu_{X^t}(a)) + \varepsilon_{a,t}$ for each agent $a \in \mathcal{I}^t \cup \mathcal{J}^t$, where the $\varepsilon_{a,t}$ are independent, 1-subgaussian random variables.
4. The platform incurs regret equal to the *instability* of the selected market outcome $(X^t, \tau^t)$.

The platform's total regret $R_T$ is thus the cumulative instability incurred up through round $T$.

### 3.1 Preference structure

In this bandits framework, we can embed varying degrees of structure on agent preferences. We capture these preference structures by the functional form of agents' utility functions and its relation to agent contexts. More formally, let $\mathcal{U}$ be the set of functions $u: \mathcal{A}^* \times \mathcal{A}^* \to \mathbb{R}$, i.e., $\mathcal{U}$ is the set of all possible (global) utility functions. We introduce our preference structures as subsets of $\mathcal{U}$.

**Unstructured preferences.** The basic setting is one where the preferences are unstructured. Specifically, we consider the class of utility functions $\mathcal{U}_{\text{unstructured}} = \{u \in \mathcal{U} \mid u(a, a') \in [-1, 1]\}$. In this setup, the platform must learn each agent's utility function $u_a(\cdot) = u(a, \cdot)$.

**Separable linear preferences.** We also consider markets where each agent has *known* information given by their context as well as *hidden* information that must be learned by the platform. We explore this setting under the assumption that agents' contexts and hidden information interact linearly.

We assume that all contexts belong to $\mathcal{B}^d$ (i.e., $\mathcal{C} = \mathcal{B}^d$) where $\mathcal{B}^d$ is the $\ell_2$ unit ball in $\mathbb{R}^d$. We also assume that there exists a function $\phi: \mathcal{A}^* \to \mathcal{B}^d$ mapping each agent to the hidden information associated to that agent. The preference class $\mathcal{U}_{\text{linear}}^d$ can then be defined as

$$\mathcal{U}_{\text{linear}}^d = \left\{ u \in \mathcal{U} \mid u(a, a') = \langle c_{a'}, \phi(a) \rangle \text{ for some } \phi: \mathcal{A}^* \to \mathcal{B}^d \right\}.$$

## 4 Measuring Approximate Stability

When learning stable matchings, we must settle for guarantees of approximate stability, since exact stability—a binary notion—is unattainable when preferences are uncertain. To see this, we return to

the example from Figure 1. Suppose that the platform has uncertainty sets given by the right panel. Recall that for the true utilities, all stable outcomes match Charlene with Percy. If the true utilities were instead the upper bounds of each uncertainty set, then all stable outcomes would match Charlene and Quinn. Given only the uncertainty sets, it is impossible for the platform to find an (exactly) stable matching, so it is necessary to introduce a measure of approximate stability as a relaxed benchmark for the platform; we turn to this now.

Given the insights of Shapley and Shubik [32]—that all stable outcomes maximize the sum of agents' utilities—it might seem natural to measure distance from stability simply in terms of the *utility difference*. Formally, let $\mathcal{A}$ be the set of agents participating in the market. (This corresponds to $\mathcal{A}^t$ at time step $t$ in the bandits model.) The utility difference[3] of a market outcome $(X, \tau)$ is

$$\left( \max_{X' \in \mathscr{X}_{\mathcal{A}}} \sum_{a \in \mathcal{A}} u_a(\mu_{X'}(a)) \right) - \left( \sum_{a \in \mathcal{A}} u_a(\mu_X(a)) + \tau_a \right). \tag{3}$$

The first term is the maximum total utility of any matching, and the second term is the total utility of market outcome $(X, \tau)$. Since transfers are zero-sum, (3) can be equivalently written as

$$\left( \max_{X' \in \mathscr{X}_{\mathcal{A}}} \sum_{a \in \mathcal{A}} u_a(\mu_{X'}(a)) \right) - \sum_{a \in \mathcal{A}} u_a(\mu_X(a)).$$

But this shows that utility difference ignores the transfers $\tau$ entirely! In fact, the utility difference can be zero even when the transfers lead to a market outcome that is far from stable (see Appendix A.1). Utility difference is therefore *not* incentive-aware, making it unsuitable as an objective for learning stable matchings with transfers.

In the remainder of this section, we propose a measure of instability—Subset Instability—and show that it serves as a well-motivated and suitable objective for learning stable matchings with transfers.

## 4.1 Subset Instability

Subset Instability is based on utility difference, but rather than only looking at the market in aggregate, it takes a maximum ranging over all subsets of agents.

**Definition 4.1.** Given utilities $u$, the *Subset Instability* of a matching with transfers $(X, \tau)$ is

$$I(X, \tau; u, \mathcal{A}) := \max_{\mathcal{S} \subseteq \mathcal{A}} \left[ \left( \max_{X' \in \mathscr{X}_{\mathcal{S}}} \sum_{a \in \mathcal{S}} u_a(\mu_{X'}(a)) \right) - \left( \sum_{a \in \mathcal{S}} u_a(\mu_X(a)) + \tau_a \right) \right]. \tag{$*$}$$

(The first term is the maximum total utility of any matching over $\mathcal{S}$, and the second term is the total utility of the agents in $\mathcal{S}$ under market outcome $(X, \tau)$.)

Intuitively, Subset Instability captures stability because it checks whether any subset of agents would prefer an alternate outcome. We provide a more extensive economic interpretation below; but before doing so, we first illustrate Definition 4.1 in the context of the example in Figure 1.

Consider the matching $X = \{(C, Q)\}$ with transfers $\tau_C = -11$ and $\tau_Q = 11$. (This market outcome is stable for the upper bounds of the uncertainty sets of the platform in Figure 1, but not stable for the true utilities.) Note that the subset $\mathcal{S}$ that maximizes Subset Instability is $\mathcal{S} = \{C, P\}$, in which case $\max_{X' \in \mathscr{X}_{\mathcal{S}}} \sum_{a \in \mathcal{S}} u_a(\mu_{X'}(a)) = 4$ and $\sum_{a \in \mathcal{S}} (u_a(\mu_X(a)) + \tau_a) = 1$. Thus, the Subset Instability of $(X, \tau)$ is $I(X, \tau; u, \mathcal{A}) = 4 - 1 = 3$. In contrast, the utility difference of $(X, \tau)$ is 2.

We now give two interpretations of Subset Instability to provide further insight into why Subset Instability is a meaningful notion of approximate stability in online marketplaces. Specifically, we interpret Subset Instability as the *minimum stabilizing subsidy* and as a *measure of user unhappiness*.

**Subset Instability as the platform's minimum stabilizing subsidy.** Subset Instability can be interpreted in terms of monetary subsidies from the platform to the agents. Specifically, the Subset

---

[3]Utility difference is standard as a measure of regret for learning a maximum weight matching in the combinatorial bandits literature (see, e.g., [14]). However, we show that for learning stable matchings, a fundamentally different measure of regret is needed.

Instability of a market outcome equals the minimum amount the platform could subsidize agents so that the subsidized market outcome is individually rational and has no blocking pairs.

Formally, let $s \in \mathbb{R}_{\geq 0}^{\mathcal{A}}$ denote subsidies made by the platform, where $s_a \geq 0$ is the subsidy provided to agent $a$. For a market outcome $(X, \tau)$, the *minimum stabilizing subsidy* is

$$\min_{s \in \mathbb{R}_{\geq 0}^{\mathcal{A}}} \left\{ \sum_{a \in \mathcal{A}} s_a \;\middle|\; (X, \tau + s) \text{ is stable} \right\}, \tag{4}$$

where we define stability in analogy to Definition 2.1. Specifically, we say that a market outcome $(X, \tau)$ with subsidies $s$ is *stable* if it is individually rational, i.e., $u_a(\mu_X(a)) + \tau_a + s_a \geq 0$ for all agents $a \in \mathcal{A}$, and has no blocking pairs, i.e., $(u_i(\mu_X(i)) + \tau_i + s_i) + (u_j(\mu_X(j)) + \tau_j + s_j) \geq u_i(j) + u_j(i)$ for all pairs of agents $(i, j) \in \mathcal{I} \times \mathcal{J}$. We show the following equivalence:

**Proposition 4.2.** *Minimum stabilizing subsidy equals Subset Instability for any market outcome.*

The proof boils down to showing that the two definitions are "dual" to each other. To formalize this, we rewrite the minimum stabilizing subsidy as the solution to the following linear program:

$$\min_{s \in \mathbb{R}^{|\mathcal{A}|}} \sum_{a \in \mathcal{A}} s_a \tag{5}$$

$$\text{s.t. } \big(u_i(\mu_X(i)) + \tau_i + s_i\big) + \big(u_j(\mu_X(j)) + \tau_j + s_j\big) \geq u_i(j) + u_j(i) \qquad \forall (i, j) \in \mathcal{I} \times \mathcal{J}$$

$$u_a(\mu_X(a)) + \tau_a + s_a \geq 0 \qquad \forall a \in \mathcal{A}$$

$$s_a \geq 0 \qquad \forall a \in \mathcal{A}.$$

The crux of our argument is that the dual linear program to (5) maximizes the combinatorial objective $(*)$. The equivalence of $(*)$ and (5) then follows from strong duality.

**Subset Instability as a measure of user unhappiness.** While the above interpretations focus on Subset Instability from the platform's perspective, we show that Subset Instability can also be interpreted as a measure of user unhappiness. Given a subset $\mathcal{S} \subseteq \mathcal{A}$ of agents, which we call a coalition, we define the *unhappiness* of $\mathcal{S}$ with respect to a market outcome $(X, \tau)$ to be the maximum gain (relative to $(X, \tau)$) in total utility that the members of coalition $\mathcal{S}$ could achieve by matching only among themselves, such that no member is worse off than they were in $(X, \tau)$. (See Appendix A.3 for a formal definition.) The condition that no member is worse off ensures that all agents would actually want to participate in the coalition (i.e. they prefer it to the original market outcome).

User unhappiness differs from the original definition $(*)$ of Subset Instability, since $(*)$ does not require agents to be better off in any alternative matching. We show this difference is inconsequential:

**Proposition 4.3.** *The maximum unhappiness of any coalition $\mathcal{S} \subseteq \mathcal{A}$ with respect to $(X, \tau)$ equals the Subset Instability $I(X, \tau; u, \mathcal{A})$.*

The main takeaway from Proposition 4.3 is that Subset Instability not only measures costs to the platform, but also costs to users, in terms of the maximum amount they "leave on the table" by not negotiating an alternate arrangement amongst themselves.

## 4.2 Properties of Subset Instability

We now describe additional properties of our instability measure that are important for learning. We show that Subset Instability is: (i) zero if and only if the matching with transfers is stable, (ii) Lipschitz in the true utility functions, and (iii) lower bounded by the utility difference.

**Proposition 4.4.** *Subset Instability satisfies the following properties:*

1. *Subset Instability is always nonnegative and is zero if and only if $(X, \tau)$ is stable.*
2. *Subset Instability is Lipschitz continuous with respect to agent utilities. That is, for any possible market outcome $(X, \tau)$, and any pair of utility functions $u$ and $\tilde{u}$ it holds that:*

$$|I(X, \tau; u, \mathcal{A}) - I(X, \tau; \tilde{u}, \mathcal{A})| \leq 2 \sum_{a \in \mathcal{A}} \|u_a - \tilde{u}_a\|_\infty.$$

3. *Subset Instability is always at least the utility difference.*

These three properties show that Subset Instability is useful as a regret measure for learning stable matchings. The first property establishes that Subset Instability satisfies the basic desideratum of having zero instability coincide with exact stability. The second property shows Subset Instability is robust to small perturbations to agents' utility functions. The third property ensures that, when learning using Subset Instability as a loss function, the platform learns a socially optimal matching.

# 5   Learning Stable Matchings in a Bandits Framework

In this section, we develop a general approach for designing algorithms that achieve near-optimal regret within our framework. While our framework bears some resemblance to the (incentive-free) combinatorial bandit problem of learning a maximum weight matching, our setting has two crucial differences: (i) in each round, the platform must choose *transfers* in addition to a matching, and (ii) loss is measured with respect to *instability* rather than the utility difference. Nonetheless, we show that a suitable interpretation of "optimism in the face of uncertainty" can still apply.

**Regret bounds for different preference structures.**   By instantiating this optimism-based approach, we derive regret bounds for the preference structures introduced in Section 3. We start with the simplest case of unstructured preferences, where we assume no structure on the utilities.

**Theorem 5.1.** *For preference class $\mathcal{U}_{\text{unstructured}}$ (see Section 3),* MATCHUCB *(defined in Section 5.2) incurs an expected regret of $\mathbb{E}(R_T) = O\big(|\mathcal{A}|\sqrt{nT\log(|\mathcal{A}|T)}\big)$, where $n = \max_t |\mathcal{A}_t|$.*

In Section 5.3, we additionally give a matching (up to logarithmic factors) lower bound showing for $n = |\mathcal{A}|$ that such scaling in $|\mathcal{A}|$ is indeed necessary. Thus, regret must scale as $|\mathcal{A}|\sqrt{n}$, which is superlinear in the size of the market. This bound means that the platform needs to learn a superconstant amount of information per agent in the marketplace. This suggests that, without preference structure, it is unlikely that a platform can efficiently learn a stable matching in a large market.

The next bound demonstrates that, with preference structure, efficient learning of a stable matching becomes possible. Specifically, we consider separable linear preferences, where the platform needs to learn hidden information associated to each agent. We show the following:

**Theorem 5.2.** *For preference class $\mathcal{U}_{\text{linear}}$ (see Section 3),* MATCHLINUCB *(defined in Section 5.2) incurs an expected regret of $\mathbb{E}(R_T) = O\big(d\sqrt{|\mathcal{A}|}\sqrt{nT\log(|\mathcal{A}|T)}\big)$, where $n = \max_t |\mathcal{A}_t|$.*

When $n$ is comparable to $|\mathcal{A}|$, the regret bound in Theorem 5.2 scales linearly with the market size (captured by $|\mathcal{A}|$) and linearly with the dimension $d$. This means that the platform learns (at most) a constant amount of information per agent in the marketplace. We interpret this as indicating that the platform can efficiently learn a stable matching in large markets for separable linear preferences.

## 5.1   Algorithm

---

**Algorithm 1** COMPUTEMATCH: Compute matching with transfers from confidence sets

---

 1: **procedure** COMPUTEMATCH($\mathscr{C}$)
 2:     **for** $(i,j) \in \mathcal{I} \times \mathcal{J}$ **do**                     ▷ Instantiate UCB estimates of utilities.
 3:         $u_i^{\text{UCB}}(j) \leftarrow \max\big(C_{i,j}\big); u_j^{\text{UCB}}(i) \leftarrow \max\big(C_{j,i}\big)$
 4:     $(X^*, p^*) \leftarrow$ optimal primal-dual pair for (P) and (D) given utilities $u^{\text{UCB}}$
 5:     **for** $a \in \mathcal{A}$ **do**                          ▷ Set transfers based on $(X^*, p^*)$ and UCB utilities.
 6:         $\tau_a \leftarrow p_a^* - u_a^{\text{UCB}}(\mu_{X^*}(a))$
 7:     **return** $(X^*, \tau)$

---

Following the principle of optimism, our algorithm selects at each round a stable market outcome using upper confidence bounds as if they were the true agent utilities. To design and analyze this algorithm, we leverage the fact that, in the full-information setting, stable market outcomes are optimal solutions to a pair of primal-dual linear programs whose coefficients depend on agents' utility functions. This primal-dual perspective lets us compute a market outcome each round. A particular consequence is that any UCB-based algorithm for learning matchings in a semi-bandit setting can be transformed into an algorithm for learning *both* the matching and the prices.

**Stable market outcomes via linear programming duality.** Before proceeding with the details of our algorithm, we review how the primal-dual framework can be used to select a stable market outcome in the full information setting. Shapley and Shubik [32] show that stable market outcomes $(X, \tau)$ correspond to optimal primal-dual solutions to the following pair of primal and dual linear programs (where we omit the round index $t$ and consider matchings over $\mathcal{A} = \mathcal{I} \cup \mathcal{J}$):

**Primal** (P)

$$\max_{Z \in \mathbb{R}^{|\mathcal{I}| \times |\mathcal{J}|}} \sum_{(i,j) \in \mathcal{I} \times \mathcal{J}} Z_{i,j}(u_i(j) + u_j(i))$$

$$\text{s.t.} \quad \sum_{j \in \mathcal{J}} Z_{i,j} \leq 1 \quad \forall i \in \mathcal{I}$$

$$\sum_{i \in \mathcal{I}} Z_{i,j} \leq 1 \quad \forall j \in \mathcal{J}$$

$$Z_{i,j} \geq 0 \quad \forall (i,j) \in \mathcal{I} \times \mathcal{J}$$

**Dual** (D)

$$\min_{p \in \mathbb{R}^{|\mathcal{A}|}} \sum_{a \in \mathcal{A}} p_a$$

$$\text{s.t.} \quad p_i + p_j \geq u_i(j) + u_j(i) \quad \forall (i,j) \in \mathcal{I} \times \mathcal{J}$$

$$p_a \geq 0 \quad \forall a \in \mathcal{A}$$

Given any optimal primal-dual solution $(Z, p)$ to (P) and (D), one can recover a matching $\mu_X$ from the nonzero entries of $Z$ and set transfers $\tau_a = p_a - u_a(\mu_X(a))$ to obtain a stable outcome $(X, \tau)$. Moreover, any stable outcome induces an optimal primal-dual pair $(Z, p)$.

**Overview of the algorithm.** Each round, we compute a matching with transfers by solving (P) and (D) for our upper confidence bounds: Suppose that we have a collection $\mathscr{C}$ of confidence sets $C_{i,j}, C_{j,i} \subseteq \mathbb{R}$ such that $u_i(j) \in C_{i,j}$ and $u_j(i) \in C_{j,i}$ for all $(i,j) \in \mathcal{I} \times \mathcal{J}$. We use $\mathscr{C}$ to get an upper confidence bound estimate for each agent's utility function and then compute a stable matching with transfers as if these estimates were the true utilities—see Algorithm 1 (COMPUTEMATCH).

The key fact we need to analyze our algorithms is that Subset Instability is upper bounded by the sum of the sizes of the relevant confidence sets, assuming that the confidence sets contain the true utilities. We show this in the following lemma, where we again omit the round index $t$:

**Lemma 5.3.** *Suppose a collection of confidence sets $\mathscr{C}$ is such that $u_i(j) \in C_{i,j}$ and $u_j(i) \in C_{j,i}$ for all $(i,j) \in \mathcal{I} \times \mathcal{J}$. Then the instability of $(X^{\mathrm{UCB}}, \tau^{\mathrm{UCB}}) := \text{COMPUTEMATCH}(\mathscr{C})$ satisfies*

$$I(X^{\mathrm{UCB}}, \tau^{\mathrm{UCB}}; u, \mathcal{A}^t) \leq \sum_{a \in \mathcal{A}} \Big( \max\big(C_{a, \mu_{X^{\mathrm{UCB}}}(a)}\big) - \min\big(C_{a, \mu_{X^{\mathrm{UCB}}}(a)}\big) \Big). \tag{6}$$

To prove this, write $I(X^{\mathrm{UCB}}, \tau^{\mathrm{UCB}}; u, \mathcal{A}^t) = I(X^{\mathrm{UCB}}, \tau^{\mathrm{UCB}}; u, \mathcal{A}^t) - I(X^{\mathrm{UCB}}, \tau^{\mathrm{UCB}}; u^{\mathrm{UCB}}, \mathcal{A}^t)$. Now, it may be tempting to bound this difference using the Lipschitz continuity of Subset Instability. However, such a bound would depend on the sizes of the confidence sets for all pairs of agents, including those *not* matched in $X^{\mathrm{UCB}}$, making it too weak to analyze UCB-style algorithms. Thus, we use the fact that $u^{\mathrm{UCB}}$ is a pointwise upper bound on $u$ to obtain a tighter analysis.

## 5.2 Explicit algorithms

The regret bound of Lemma 5.3 suggests an algorithmic approach: each round, select the matching with transfers returned by COMPUTEMATCH and update confidence sets accordingly. To instantiate this approach, it remains to construct confidence intervals that contain the true utilities with high probability. This last step naturally depends on the assumptions made about the utilities and the noise.

**Unstructured preferences.** We construct confidence intervals following the classical UCB approach: for each utility value involving the pair $(i,j) \in \mathcal{I} \times \mathcal{J}$, we take a length $O\big(\sqrt{\log(|\mathcal{A}|T)/n_{ij}}\big)$ confidence interval centered around the empirical mean, where $n_{ij}$ is the number of times the pair has been matched so far. We give this construction precisely in Algorithm 2 (MATCHUCB).

To analyze MATCHUCB, recall that Lemma 5.3 bounds the regret at each step by the lengths of the confidence intervals of each pair in the selected matching. Bounding the lengths of the confidence intervals parallels the analysis of UCB for classical stochastic multi-armed bandits.

**Separable Linear Preferences.** We build confidence sets using the high-level idea from LinUCB [29, 22]: we compute a confidence set for each hidden vector $\phi(a)$ using a least squares estimate and use that to get confidence sets $C_{a,a'}$ for the utilities. We detail this precisely in Appendix B.3.

---

**Algorithm 2** MATCHUCB: A bandit algorithm for matching with transferable utilities for unstructured preferences.

---

1: **procedure** MATCHUCB($T$)
2:     **for** $(i, j) \in \mathcal{I} \times \mathcal{J}$ **do**                ▷ Initialize confidence intervals.
3:         $C_{i,j} \leftarrow [-1, 1]; C_{j,i} \leftarrow [-1, 1]$
4:     **for** $1 \le t \le T$ **do**
5:         $(X^t, \tau^t) \leftarrow$ COMPUTEMATCH($\mathscr{C}$)
6:         **for** $(i, j) \in X^t$ **do**           ▷ Set confidence intervals and update means.
7:             Update empirical means $\hat{u}_i(j)$ and $\hat{u}_j(i)$ from feedback; increment counter $n_{ij}$.
8:             $C_{i,j} \leftarrow \left[\hat{u}_i(j) - 8\sqrt{\log(|\mathcal{A}|T)/n_{ij}}, \hat{u}_i(j) + 8\sqrt{\log(|\mathcal{A}|T)/n_{ij}}\right] \cap [-1, 1]$
9:             $C_{j,i} \leftarrow \left[\hat{u}_j(i) - 8\sqrt{\log(|\mathcal{A}|T)/n_{ij}}, \hat{u}_j(i) + 8\sqrt{\log(|\mathcal{A}|T)/n_{ij}}\right] \cap [-1, 1]$

---

### 5.3 Matching lower bound

For the case of unstructured preferences, we now show MATCHUCB achieves optimal regret (up to logarithmic factors) by showing a lower bound that (nearly) matches the upper bound in Theorem 5.1.

**Lemma 5.4.** *Any algorithm that learns a stable matching with respect to unstructured preferences has an instance on which it has expected regret $\widetilde{\Omega}(|A|^{3/2}\sqrt{T})$ (as measured by Subset Instability).*

To show the lower bound, we prove a lower bound for the *easier* (by Proposition 4.4) problem of learning a maximum weight matching. This lower bound illustrates the close connection between our setting and that of learning a maximum weight matching. Indeed, by applying MATCHUCB and simply disregarding the transfers every round, we recover the classical UCB-based algorithm for learning the maximum weight matching [14, 9, 21]. From this perspective, the contribution of MATCHUCB is an approach to set the dual variables while asymptotically maintaining the same regret as the primal-only problem.

## 6 Extensions

**Instance-dependent regret bounds**     While we have focused on bounds that hold uniformly for all problem instances, we now explore *instance-dependent* regret bounds. Instance-dependent bounds capture a different facet of bandit algorithms: how does the number of mistakes made by the algorithm on each instance scale with respect to $T$? We show $O(\log T)$ regret bounds in the full version [18].

**Matching with non-transferable utilities**     While we have focused on matching with transferable utilities, utilities are not always transferable with money in practice (e.g., dating markets and college admissions). In the full version [18], we extend our findings to the matching with non-transferable utilities (NTU) model [15], which has also been studied in a learning context [11, 24, 7, 30]. The definition of Subset Instability extends naturally and has advantages over the "utility difference" metric used in prior work. We further show that a suitable interpretation of UCB can also be applied to this setting.

## 7 Discussion

We introduced a framework for learning equilibria in matching markets from bandit feedback. A core component is Subset Instability, which captures the distance of a market outcome from equilibrium. Using Subset Instability as a loss function, we designed UCB-based learning algorithms that optimize for alignment with agent incentives. The resulting regret bounds for different preference structures take a step towards elucidating when a marketplace can efficiently learn stable outcomes.

While we focused on matching markets with stochastic bandit feedback in this work, we believe that the question of learning equilibria in data-driven marketplaces is of much more general interest. An interesting direction for future inquiry would be to explore in what other settings, and with what other algorithmic methods, can equilibria be learned?[4]

---

[4]For a more detailed discussion of directions for future work, see the full version of the paper [18].

## Acknowledgments and Disclosure of Funding

We would like to thank Itai Ashlagi, Jiantao Jiao, Scott Kominers (along with the Lab for Economic Design), Cassidy Laidlaw, Celestine Mendler-Dünner, and Banghua Zhu for valuable feedback. Meena Jagadeesan acknowledges support from the Paul and Daisy Soros Fellowship; Alexander Wei acknowledges support from an NSF Graduate Research Fellowship under Grant No. 1752814.

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
