# A Proofs for Section 4

This section contains further exposition (including proofs) for Section 4.

## A.1 Limitations of utility difference as an instability measure

To illustrate why utility difference fails to be a good measure of instability, we describe a matching with transfers that (i) is far from stable and (ii) has $0$ utility difference (but large Subset Instability).

**Example A.1.** Consider the following market with two agents: $\mathcal{I} = \{i\}$ and $\mathcal{J} = \{j\}$. Suppose that $u_i(j) = 2$ and $u_j(i) = -1$. Consider the matching $X = \{(i, j)\}$ with transfers $\tau_i = -\xi$ and $\tau_j = \xi$ for some $\xi > 0$. We will show that this matching with transfers will have the properties stated above when $\xi$ is large.

This matching with transfers has utility difference $0$ (for any $\xi$) since it maximizes the sum of utilities. Indeed, it is stable for any $\xi \in [1, 2]$. However, when $\xi > 2$, this matching with transfers is no longer stable, since the individual rationality condition $u_i(j) + \tau_i \geq 0$ fails. (Intuitively, the larger $\xi$ is, the further we are from stability.) But its utility difference remains at $0$.

On the other hand, the Subset Instability of this matching with transfers is $\xi - 2 > 0$ when $\xi > 2$. In particular, Subset Instability increases with $\xi$ in this regime, which is consistent with the intuition that outcomes with larger $\xi$ should be more unstable.

## A.2 Proof of Proposition 4.2

**Proposition 4.2.** *Minimum stabilizing subsidy equals Subset Instability for any market outcome.*

*Proof of Proposition 4.2.* We can take the dual of the linear program (5) to obtain:

$$\max_{\substack{S \in \mathbb{R}^{|\mathcal{I}| \times |\mathcal{J}|} \\ Z \in \mathbb{R}^{|\mathcal{A}|}}} \quad \sum_{(i,j) \in \mathcal{I} \times \mathcal{J}} S_{i,j} \Big( \big(u_i(j) - u_i(\mu_X(i)) - \tau_i\big) + \big(u_j(i) - u_j(\mu_X(j)) - \tau_j\big) \Big) \tag{7}$$
$$- \sum_{a \in \mathcal{A}} Z_a(u_a(\mu_X(a)) + \tau_a)$$

$$\text{s.t. } Z_i + \sum_{j \in \mathcal{J}} S_{i,j} \leq 1 \qquad \forall i \in \mathcal{I}; \qquad Z_j + \sum_{i \in \mathcal{I}} S_{i,j} \leq 1 \qquad \forall j \in \mathcal{J};$$
$$S_{i,j} \geq 0 \qquad \forall(i,j) \in \mathcal{I} \times \mathcal{J}; \qquad Z_a \geq 0 \qquad \forall a \in \mathcal{A}. \qquad .$$

By strong duality, the optimal values of (5) and (7) are equal. Thus, it suffices to show that Subset Instability is equal to (7). By Proposition 4.3, we know that Subset Instability is equal to the maximum unhappiness of any coalition. Thus, it suffices to show that (7) is equal to the maximum unhappiness of any coalition.

To interpret (7), observe that there exist optimal $S^*$ and $Z^*$ all of whose entries lie in $\{0, 1\}$ because this linear program can be embedded into a maximum weight matching linear program. Take such a choice of optimal $S^*$ and $Z^*$. Then, $S^*$ is an indicator vector corresponding to a (partial) matching on a subset of the agents such that all pairs in this matching are blocking with respect to $(X, \tau)$. Similarly, $Z^*$ is an indicator vector of agents who would rather be unmatched than match according to $(X, \tau)$.

We first prove the claim that $I(X, \tau; u, \mathcal{A})$ is at least (7). Based on the above discussion, the optimal objective of (7) is obtained through $S^*$ and $Z^*$ that represent a matching and a subset of agents respectively. Let $S$ be the union of agents participating in $S^*$ and $Z^*$. We see that the objective of (7) is equal to the utility difference at $S$, i.e.:

$$\left( \max_{X' \in \mathscr{X}_S} \sum_{a \in S} u_a(\mu_{X'}(a)) \right) - \left( \sum_{a \in S} u_a(\mu_X(a)) + \tau_a \right).$$

This is no larger than Subset Instability by definition.

We next prove the claim that $I(X, \tau; u, \mathcal{A})$ is at most (7). Let's consider $S^*$ that maximizes:

$$\max_{S \subseteq \mathcal{A}} \left( \max_{X' \in \mathscr{X}_S} \sum_{a \in S} u_a(\mu_{X'}(a)) \right) - \left( \sum_{a \in S} u_a(\mu_X(a)) + \tau_a \right).$$

Let's take the maximum weight matching of $S^*$. Let $S$ be given by the matched agents in this matching and let $Z$ be given by the unmatched agents in this matching (using the interpretation of (7) described above). We see that the objective at (7) for $(S, Z)$ is equal to Subset Instability which proves the desired statement.

$\square$

### A.3 Proof of Proposition 4.3

We first formally define the *unhappiness of a coalition*, as follows. In particular, the unhappiness with respect to $(X, \tau)$ of a coalition $\mathcal{S} \subseteq \mathcal{A}$ is defined to be:

$$\sup_{\substack{X' \in \mathscr{X}_S \\ \tau' \in \mathbb{R}^{|S|}}} \sum_{a \in \mathcal{S}} \left( u_a(\mu_{X'}(a)) + \tau_a' \right) - \sum_{a \in \mathcal{S}} \left( u_a(\mu_X(a)) + \tau_a \right) \qquad (8)$$

$$\text{s.t.} \quad u_a(\mu_{X'}(a)) + \tau_a' \geq u_a(\mu_X(a)) + \tau_a \qquad \forall a \in \mathcal{S}$$

$$\tau_a' + \tau_{\mu_{X'}(a)}' = 0 \qquad \forall a \in \mathcal{S},$$

with unhappiness being 0 if there are no feasible $X'$ and $\tau'$. In the optimization program, $(X', \tau')$ represents a matching with transfers over $\mathcal{S}$, with the constraint $\tau_a' + \tau_{\mu_{X'}(a)}' = 0$ ensuring that it is zero-sum. The objective measures the difference between $(X, \tau)$ and $(X', \tau')$ of the total utility of the agents in $\mathcal{S}$. The constraint $u_a(\mu_{X'}(a)) + \tau_a' \geq u_a(\mu_X(a)) + \tau_a$ encodes the requirement that all agents be at least as well off under $(X', \tau')$ as they were under $(X, \tau)$. This optimization program therefore captures the objective of $\mathcal{S}$ to maximize their total payoff while ensuring that no member of the coalition is worse off than they were according to $(X, \tau)$.

Recall that, in terms of unhappiness, Proposition 4.3 is as follows:

**Proposition 4.3.** *The maximum unhappiness of any coalition $\mathcal{S} \subseteq \mathcal{A}$ with respect to $(X, \tau)$ equals the Subset Instability $I(X, \tau; u, \mathcal{A})$.*

*Proof of Proposition 4.3.* By Proposition 4.2, we know that Subset Instability is equal to (5). Moreover, by strong duality, we know that Subset Instability is equal to (7) (the dual linear program of (5)). Thus, it suffices to prove that the maximum unhappiness of any coalition is equal to (7).

We first prove the claim that (7) is at most the maximum unhappiness of any coalition with respect to $(X, \tau)$. To do this, it suffices to construct a coalition $\mathcal{S} \subseteq \mathcal{A}$ such that (7) is at most the unhappiness of $\mathcal{S}$. We construct $\mathcal{S}$ as follows: Recall that there exist optimal solutions $S^*$ and $Z^*$ to (7) such that $S^*$ corresponds to a (partial) matching on $\mathcal{I} \times \mathcal{J}$ and $Z^*$ corresponds to a subset of $\mathcal{A}$. We may take $\mathcal{S}$ to be the union of the agents involved in $S^*$ and in $Z^*$. Now, we upper bound the unhappiness of $\mathcal{S}$ by constructing $X'$ and $\tau'$ that are feasible for (8). We can take $X'$ to be the matching that corresponds to the indicator vector $S^*$. Because $(S^*, Z^*)$ is optimal for (7),

$$u_i(j) + u_j(i) \geq (u_i(\mu_X(i)) + \tau_i) + (u_j(\mu_X(j)) + \tau_j)$$

for all $(i, j) \in X'$. Thus, we can find a vector $\tau'$ of transfers that is feasible for (8). Then, since $\sum_{a \in \mathcal{S}} \tau_a' = 0$, the objective of (8) at $(X', \tau')$ is

$$\sum_{a \in \mathcal{S}} \left( u_a(\mu_{X'}(a)) - u_a(\mu_X(a)) - \tau_a \right).$$

This equals to the objective of (7) at $(S^*, Z^*)$, which equals (7), as desired.

We now show the inequality in the other direction, that (7) is at least the maximum unhappiness of any coalition with respect to $(X, \tau)$. It suffices to construct a feasible solution $(S, Z)$ to (7) that achieves at least the maximum unhappiness of any coalition. Let $\mathcal{S}$ be a coalition with maximum unhappiness, and let $(X', \tau')$ be an optimal solution for (8). Moreover, let $S$ be the indicator vector

corresponding to agents who are matched in $X'$ and $Z$ be the indicator vector corresponding to agents in $\mathcal{S}$ who are unmatched. The objective of (8) at $(X', \tau')$ is

$$\sum_{a \in \mathcal{S}} \big( u_a(\mu_{X'}(a)) - u_a(\mu_X(a)) - \tau_a \big),$$

which equals the objective of (7) at the $(S, Z)$ that we constructed. $\qquad \square$

### A.4   Proof of Proposition 4.4

**Proposition 4.4.** *Subset Instability satisfies the following properties:*

1. *Subset Instability is always nonnegative and is zero if and only if $(X, \tau)$ is stable.*
2. *Subset Instability is Lipschitz continuous with respect to agent utilities. That is, for any possible market outcome $(X, \tau)$, and any pair of utility functions $u$ and $\tilde{u}$ it holds that:*

$$|I(X, \tau; u, \mathcal{A}) - I(X, \tau; \tilde{u}, \mathcal{A})| \leq 2 \sum_{a \in \mathcal{A}} \|u_a - \tilde{u}_a\|_\infty.$$

3. *Subset Instability is always at least the utility difference.*

*Proof of Proposition 4.4.* We first prove the third part of the Proposition statement, then the first part of the Proposition statement, and finally the second part.

**Proof of part (c).**   Because $\sum_{a \in \mathcal{A}} \tau_a = 0$, Subset Instability satisfies the following:

$$
\begin{aligned}
I(X, \tau; u, \mathcal{A}) &\geq \left( \max_{X' \in \mathscr{X}_\mathcal{A}} \sum_{a \in \mathcal{A}} u_a(\mu_{X'}(a)) \right) - \left( \sum_{a \in \mathcal{A}} u_a(\mu_X(a)) + \tau_a \right) \\
&= \left( \max_{X' \in \mathscr{X}_\mathcal{A}} \sum_{a \in \mathcal{A}} u_a(\mu_{X'}(a)) \right) - \left( \sum_{a \in \mathcal{A}} u_a(\mu_X(a)) \right).
\end{aligned}
$$

The second line is exactly the utility difference.

**Proof of part (a).**   From above, we have that Subset Instability is lower bounded by the utility difference, which is always nonnegative. Hence Subset Instability is also always nonnegative.

To see that Subset Instability is $0$ if and only if $(X, \tau)$ is stable, first suppose $(X, \tau)$ is unstable. Then, there exists a blocking pair $(i, j)$, in which case

$$I(X, \tau; u, \mathcal{A}) \geq u_i(j) + u_j(i) - (u_i(\mu_X(i)) + u_j(\mu_X(j)) + \tau_i + \tau_j) > 0$$

by the definition of blocking. Now, suppose $I(X, \tau; u, \mathcal{A}) > 0$. Then, there exists a subset $S \subseteq \mathcal{A}$ such that

$$\left( \max_{X' \in \mathscr{X}_S} \sum_{a \in S} u_a(\mu_{X'}(a)) \right) - \left( \sum_{a \in S} u_a(\mu_X(a)) + \tau_a \right) > 0.$$

Let $X'$ be a maximum weight matching on $S$. We can rewrite the above as

$$\sum_{(i,j) \in X'} \big( u_i(j) + u_j(i) - (u_i(\mu_X(i)) + u_j(\mu_X(j)) + \tau_i + \tau_j) \big) > 0.$$

Some term in the sum on the left-hand side must be positive, so there exists a blocking pair $(i, j) \in X'$. In particular, $(X, \tau)$ is not stable.

**Proof of part (b).** We prove that

$$|I(X, \tau; u, \mathcal{A}) - I(X, \tau; \tilde{u}, \mathcal{A})| \leq 2 \sum_{a \in \mathcal{A}} \|u_a - \tilde{u}_a\|_\infty.$$

The supremum of $L$-Lipschitz functions is $L$-Lipschitz, so it suffices to show that

$$\left( \max_{X' \in \mathscr{X}_S} \sum_{a \in S} u_a(\mu_{X'}(a)) \right) - \sum_{a \in S} (u_a(\mu_X(a)) + \tau_a)$$

satisfies the desired Lipschitz condition for any $S \subseteq \mathcal{A}$. In particular, it suffices to show that

$$\left| \sum_{a \in S} (u_a(\mu_X(a)) + \tau_a) - \sum_{a \in S} (\tilde{u}_a(\mu_X(a)) + \tau_a) \right| \leq \sum_{a \in \mathcal{A}} \|u_a - \tilde{u}_a\|_\infty \tag{9}$$

and

$$\left| \left( \max_{X' \in \mathscr{X}_S} \sum_{a \in S} u_a(\mu_{X'}(a)) \right) - \left( \max_{X' \in \mathscr{X}_S} \sum_{a \in S} \tilde{u}_a(\mu_{X'}(a)) \right) \right| \leq \sum_{a \in \mathcal{A}} \|u_a - \tilde{u}_a\|_\infty. \tag{10}$$

For (9), we have

$$\left| \sum_{a \in S} (u_a(\mu_X(a)) + \tau_a) - \sum_{a \in S} (\tilde{u}_a(\mu_X(a)) + \tau_a) \right| = \left| \sum_{a \in S} \big( u_a(\mu_X(a)) - \tilde{u}_a(\mu_X(a)) \big) \right|$$

$$\leq \sum_{a \in \mathcal{A}} \|u_a - \tilde{u}_a\|_\infty.$$

For (10), this boils down to showing that the total utility of the maximum weight matching is Lipschitz. Using again the fact that the supremum of Lipschitz functions is Lipschitz, this follows from the total utility of any fixed matching being Lipschitz. $\square$

# B   Proofs for Section 5

## B.1   Proof of Lemma 5.3

**Lemma 5.3.** *Suppose a collection of confidence sets $\mathscr{C}$ is such that $u_i(j) \in C_{i,j}$ and $u_j(i) \in C_{j,i}$ for all $(i, j) \in \mathcal{I} \times \mathcal{J}$. Then the instability of $(X^{\mathrm{UCB}}, \tau^{\mathrm{UCB}}) := \mathrm{COMPUTEMATCH}(\mathscr{C})$ satisfies*

$$I(X^{\mathrm{UCB}}, \tau^{\mathrm{UCB}}; u, \mathcal{A}^t) \leq \sum_{a \in \mathcal{A}} \Big( \max\big(C_{a, \mu_{X^{\mathrm{UCB}}}(a)}\big) - \min\big(C_{a, \mu_{X^{\mathrm{UCB}}}(a)}\big) \Big). \tag{6}$$

*Proof of Lemma 5.3.* Because $(X^{\mathrm{UCB}}, \tau^{\mathrm{UCB}})$ is stable with respect to the UCB utilities $u^{\mathrm{UCB}}$, we have that $I(X^{\mathrm{UCB}}, \tau^{\mathrm{UCB}}; u^{\mathrm{UCB}}, \mathcal{A}^t) = 0$. Thus, bounding $I(X^{\mathrm{UCB}}, \tau^{\mathrm{UCB}}; u, \mathcal{A}^t)$ is equivalent to bounding the difference $I(X^{\mathrm{UCB}}, \tau^{\mathrm{UCB}}; u, \mathcal{A}^t) - I(X^{\mathrm{UCB}}, \tau^{\mathrm{UCB}}; u^{\mathrm{UCB}}, \mathcal{A}^t)$.

At this stage, it might be tempting to bound this difference using the Lipschitz continuity of Subset Instability (see Proposition 4.4). However, this would only allow us to obtain an upper bound of the form $\sum_{a \in \mathcal{A}} \max_{a' \in \mathcal{A}} \big( \max(C_{a,a'}) - \min(C_{a,a'}) \big)$. The problem with this bound is that it depends on the sizes of the confidence sets for all pairs of agents, including those that are *not* matched in $X^{\mathrm{UCB}}$, making it too weak to prove regret bounds for UCB-style algorithms.[5] Thus, we proceed with a more fine-grained analysis.

Define the function

$$f(\mathcal{S}, X, \tau; u) = \left( \max_{X' \in \mathscr{X}_\mathcal{S}} \sum_{a \in \mathcal{S}} u_a(\mu_{X'}(a)) \right) - \left( \sum_{a \in \mathcal{S}} u_a(\mu_X(a)) + \tau_a \right).$$

---

[5]For intuition, consider the classical stochastic multi-armed bandits setting and suppose that we could only guarantee that the loss incurred by an arm is bounded by the maximum of the sizes of the confidence sets over *all* arms. Then, we would only be able to obtain a weak bound on regret, since low-reward arms with large confidence sets may never be pulled.

By definition, $I(X, \tau; u, \mathcal{A}) = \max_{\mathcal{S} \subseteq \mathcal{A}} f(\mathcal{S}, X, \tau; u)$. It follows that

$$I(X^{\mathrm{UCB}}, \tau^{\mathrm{UCB}}; u, \mathcal{A}^t) - I(X^{\mathrm{UCB}}, \tau^{\mathrm{UCB}}; u^{\mathrm{UCB}}, \mathcal{A}^t)$$
$$\leq \max_{\mathcal{S} \subseteq \mathcal{A}} \left( f(\mathcal{S}, X^{\mathrm{UCB}}, \tau^{\mathrm{UCB}}; u) - f(\mathcal{S}, X^{\mathrm{UCB}}, \tau^{\mathrm{UCB}}; u^{\mathrm{UCB}}) \right).$$

To finish, we upper bound $f(\mathcal{S}, X^{\mathrm{UCB}}, \tau^{\mathrm{UCB}}; u) - f(\mathcal{S}, X^{\mathrm{UCB}}, \tau^{\mathrm{UCB}}; u^{\mathrm{UCB}})$ for each $\mathcal{S} \subseteq \mathcal{A}$. We decompose this expression into two terms:

$$f(\mathcal{S}, X^{\mathrm{UCB}}, \tau^{\mathrm{UCB}}; u) - f(\mathcal{S}, X^{\mathrm{UCB}}, \tau^{\mathrm{UCB}}; u^{\mathrm{UCB}})$$
$$= \underbrace{\left( \max_{X' \in \mathscr{X}_{\mathcal{S}}} \sum_{a \in \mathcal{S}} u_a(\mu_{X'}(a)) - \max_{X' \in \mathscr{X}_{\mathcal{S}}} \sum_{a \in \mathcal{S}} u_a^{\mathrm{UCB}}(\mu_{X'}(a)) \right)}_{(A)}$$
$$+ \underbrace{\left( \sum_{a \in \mathcal{S}} (u_a^{\mathrm{UCB}}(\mu_{X^{\mathrm{UCB}}}(a)) + \tau_a^{\mathrm{UCB}}) - \sum_{a \in \mathcal{S}} (u_a(\mu_{X^{\mathrm{UCB}}}(a)) + \tau_a^{\mathrm{UCB}}) \right)}_{(B)}.$$

To see that (A) is nonpositive, observe that the maximum weight matching of $\mathcal{S}$ with respect to $u$ is no larger than the maximum weight matching of $\mathcal{S}$ with respect to $u^{\mathrm{UCB}}$, since $u^{\mathrm{UCB}}$ pointwise upper bounds $u$. To upper bound (B), observe that the transfers cancel out, so the expression is equivalent to

$$\sum_{a \in \mathcal{S}} (u_a^{\mathrm{UCB}}(\mu_{X^{\mathrm{UCB}}}(a)) - u_a(\mu_{X^{\mathrm{UCB}}}(a))) \leq \sum_{a \in \mathcal{A}} \left( \max(C_{a, \mu_{X^{\mathrm{UCB}}}(a)}) - \min(C_{a, \mu_{X^{\mathrm{UCB}}}(a)}) \right). \quad \square$$

## B.2  Proof of Theorem 5.1

**Theorem 5.1.** *For preference class $\mathcal{U}_{\mathrm{unstructured}}$ (see Section 3), MATCHUCB (defined in Section 5.2) incurs an expected regret of $\mathbb{E}(R_T) = O\big(|\mathcal{A}| \sqrt{nT \log(|\mathcal{A}|T)}\big)$, where $n = \max_t |\mathcal{A}_t|$.*

*Proof of Theorem 5.1.* The starting point for our proof of Theorem 5.1 is the typical approach in multi-armed bandits and combinatorial bandits [14, 9, 22] of bounding regret in terms of the sizes of the confidence interval of the chosen arms. However, rather than using the sizes of confidence intervals to bound the utility difference (as in the incentive-free maximum weight matching setting), we bound Subset Instability through Lemma 5.3. From here on, our approach composes cleanly with existing bandits analyses; in particular, we can follow the typical combinatorial bandits approach [14, 9] to get the desired upper bound.

For completeness, we present the full proof. We divide into two cases, based on the event $E$ that all of the confidence sets contain their respective true utilities at every time step $t \leq T$. That is, $u_i(j) \in C_{i,j}$ and $u_j(i) \in C_{j,i}$ for all $(i, j) \in \mathcal{I} \times \mathcal{J}$ at all $t$.

**Case 1: $E$ holds.**  By Lemma 5.3, we may bound

$$I(X^t, \tau^t; u, \mathcal{A}^t) \leq \sum_{a \in \mathcal{A}^t} \left( \max(C_{a, \mu_{X^t}(a)}) - \min(C_{a, \mu_{X^t}(a)}) \right) = O\left( \sum_{(i,j) \in X^t} \sqrt{\frac{\log(|\mathcal{A}|T)}{n_{ij}^t}} \right),$$

where $n_{ij}^t$ is the number of times that the pair $(i, j)$ has been matched at the start of round $t$. Let $w_{i,j}^t = \frac{1}{\sqrt{n_{ij}^t}}$ be the size of the confidence set (with the log factor scaled out) for $(i, j)$ at the start of round $t$.

At each time step $t$, let's consider the list consisting of $w_{i_t, j_t}^t$ for all $(i_t, j_t) \in X^t$. Let's now consider the overall list consisting of the concatenation of all of these lists over all rounds. Let's order this list in decreasing order to obtain a list $\tilde{w}_1, \ldots, \tilde{w}_L$ where $L = \sum_{t=1}^{T} |X^t| \leq nT$. In this notation, we observe that:

$$\sum_{t=1}^{T} I(X^t, \tau^t; u, \mathcal{A}) \leq \sum_{t=1}^{T} \sum_{a \in \mathcal{A}^t} \left( \max(C_{a, \mu_{X^t}(a)}) - \min(C_{a, \mu_{X^t}(a)}) \right) = \log(|\mathcal{A}|T) \sum_{l=1}^{L} \tilde{w}_l.$$

We claim that $\tilde{w}_l \leq O\left(\min(1, \frac{1}{\sqrt{(l/|\mathcal{A}|^2)-1}})\right)$. The number of rounds that a pair of agents can have their confidence set have size at least $\tilde{w}_l$ is upper bounded by $1 + \frac{1}{\tilde{w}_l^2}$. Thus, the total number of times that any confidence set can have size at least $\tilde{w}_l$ is upper bounded by $(|\mathcal{A}|^2)(1 + \frac{1}{\tilde{w}_l^2})$.

Putting this together, we see that:

$$
\begin{aligned}
\log(|\mathcal{A}|T) \sum_{l=1}^{L} \tilde{w}_l &\leq O\left(\sum_{l=1}^{L} \min(1, \frac{1}{\sqrt{(l/|\mathcal{A}|^2)-1}})\right) \\
&\leq O\left(\log(|\mathcal{A}|T) \sum_{l=1}^{nT} \min(1, \frac{1}{\sqrt{(l/|\mathcal{A}|^2)-1}})\right) \\
&\leq O\left(|\mathcal{A}|\sqrt{nT}\log(|\mathcal{A}|T)\right).
\end{aligned}
$$

**Case 2: $E$ does not hold.** Since each $n_{ij}(\hat{u}_i(j) - u_i(j))$ is mean-zero and 1-subgaussian, and we have $O(|\mathcal{I}||\mathcal{J}|T)$ such random variables over $T$ rounds, the probability that any of them exceeds

$$2\sqrt{\log(|\mathcal{I}||\mathcal{J}|T/\delta)} \leq 2\sqrt{\log(|\mathcal{A}|^2T/\delta)}$$

is at most $\delta$ by a standard tail bound for the maximum of subgaussian random variables. It follows that $E$ fails to hold with probability at most $|\mathcal{A}|^{-2}T^{-2}$. In the case that $E$ fails to hold, our regret in any given round would be at most $4|\mathcal{A}|$ by the Lipschitz property in Proposition 4.4. (Recall that our upper confidence bound for any utility is wrong by at most 2 due to clipping each confidence interval to lie in $[-1, 1]$.) Thus, the expected regret from this scenario is at most

$$|\mathcal{A}|^{-2}T^{-2} \cdot 4|\mathcal{A}|T \leq 4|\mathcal{A}|^{-1}T^{-1},$$

which is negligible compared to the regret bound from when $E$ does occur. $\qquad\square$

### B.3 Proof of Theorem 5.2

We describe MATCHLINUCB more formally. Let $\mathcal{T}_a$ be the set of rounds where agent $a$ is matched on the platform thus far, and for $t' \in \mathcal{T}_a$, let $\mathcal{R}_{a,t'}$ be the observed utility at time $t'$ for agent $a$. The center of the confidence set will be given by the least squares estimate

$$\phi^{\mathrm{LS}}(a) = \arg\min_{v \in \mathcal{B}^d} \left(\sum_{t' \in \mathcal{T}_a} \left(\langle v, c_{\mu_{X_{t'}}(a)}\rangle - \mathcal{R}_{a,t'}\right)\right).$$

The confidence set for $\phi(a)$ is given by

$$C_{\phi(a)} := \left\{v \;\middle|\; \sum_{t' \in \mathcal{T}_{a,t}} \left\langle v - \phi^{\mathrm{LS}}(a), c_{\mu_{X_{t'}}(a)}\right\rangle^2 \leq \beta \text{ and } \|v\|_2 \leq 1\right\},$$

where $\beta = O\left(D \log T + \frac{n_a \sqrt{\ln(n_a/\delta)}}{T^2}\right)$ and $n_a$ counts the number of times that $a$ has appeared in selected matchings. The confidence set for $u(a, a')$ is given by

$$C_{a,a'} := \left\{\langle c_{a'}, v\rangle \;\middle|\; v \in C_{\phi(a)}\right\} \cap [-1, 1].$$

**Theorem 5.2.** *For preference class $\mathcal{U}_{\mathrm{linear}}$ (see Section 3),* MATCHLINUCB *(defined in Section 5.2) incurs an expected regret of $\mathbb{E}(R_T) = O\left(d\sqrt{|\mathcal{A}|}\sqrt{nT \log(|\mathcal{A}|T)}\right)$, where $n = \max_t |\mathcal{A}_t|$.*

To prove Theorem 5.2, it suffices to (a) show that the confidence sets contain the true utilities with high probability, and (b) bound the sum of the sizes of the confidence sets.

Part (a) follows from fact established in existing analysis of LinUCB in the classical linear contextual bandits setting [29].

**Algorithm 3** MATCHLINUCB: A bandit algorithm for matching with transferable utilities for separable linear preferences.

---
1: **procedure** MATCHLINUCB($T$)
2:     **for** $(i,j) \in \mathcal{I} \times \mathcal{J}$ **do**                          ▷ Initialize confidence intervals.
3:         $C_{i,j} \leftarrow [-1, 1]$
4:         $C_{j,i} \leftarrow [-1, 1]$
5:     **for** $1 \leq t \leq T$ **do**
6:         $(X^t, \tau^t) \leftarrow$ COMPUTEMATCH($\mathscr{C}$)
7:         **for** $a \in \mathcal{A}^t$ **do**                      ▷ Update confidence intervals.
8:             Increment the counter $n_a$.
9:             $\beta \leftarrow O\left(d\log T + \frac{n_a\sqrt{\ln(n_a/(T|A|))}}{T^2}\right).$     ▷ Parameter for width of confidence set.
10:             **if** $\mu_{X^t}(a) \neq a$ **then**
11:                 Add $t$ to $\mathcal{T}_a$ (the set of rounds in which agent $a$ has been matched).
12:                 Set $\mathcal{R}_{a,t}$ equal to the observed utility for agent $a$ in round $t$.
13:                 $\phi^{\mathrm{LS}}(a) \leftarrow \operatorname{argmin}_{v \in \mathcal{B}^d}\left(\sum_{t' \in \mathcal{T}_a}\left(\langle v, c_{\mu_{X_{t'}}(a)}\rangle - \mathcal{R}_{a,t'}\right)^2\right)$     ▷ Least squares estimate.
14:                 $C_{\phi(a)} \leftarrow \left\{v \mid \sum_{t' \in \mathcal{T}_a}\left(\langle v - \phi^{\mathrm{LS}}(a), c_{\mu_{X_{t'}}(a)}\rangle\right)^2 \leq \beta, \|v\|_2 \leq 1\right\}$     ▷ Conf. ellipsoid.
15:                 **for** $a' \in \mathcal{A}$ **do**
16:                     $C_{a,a'} \leftarrow \left\{\langle c_{a'}, v\rangle \mid v \in C_{\phi(a)}\right\} \cap [-1, 1].$     ▷ Update confidence sets involving $a$.

---

**Lemma B.1** ([29, Proposition 2]). *Let the confidence sets be defined as above (and in* MATCHLINUCB*). For each $a \in \mathcal{A}$, it holds that:*

$$\mathbb{P}[\phi(a) \in C_{\phi(a)} \quad \forall 1 \leq t \leq T] \geq 1 - 1/(|\mathcal{A}|^3 T^2).$$

**Lemma B.2.** *Let the confidence sets be defined as above (and in* MATCHLINUCB*). For each $a \in \mathcal{A}$ and for any $\varepsilon > 0$, it holds that:*

$$\sum_{t|a \in \mathcal{A}^t, \mu_{X^t}(a) \neq a} \mathbf{1}\left[\max\left(C_{a,\mu_{X^t}(a)}\right) - \min\left(C_{a,\mu_{X^t}(a)}\right)\right) > \varepsilon] \leq O\left(\left(\frac{4\beta_T}{\varepsilon^2} + 1\right) d\log(1/\varepsilon)\right).$$

*Proof.* We follow the same argument as the proof of Proposition 3 in [29].

We first recall the definition of $\varepsilon$-dependence and $\varepsilon$-eluder dimension: We say that an agent $a'$ is $\varepsilon$-*dependent* on $a'_1, \ldots, a'_s$ if for all $\phi(a), \tilde{\phi}(a) \in \mathcal{B}^d$ such that

$$\sum_{k=1}^{s} \langle c_{a'_k}, \tilde{\phi}(a) - \phi(a)\rangle^2 \leq \varepsilon^2,$$

we also have $\langle c_{a'}, \tilde{\phi}(a) - \phi(a)\rangle^2 \leq \varepsilon^2$. The $\varepsilon$-eluder dimension $d_{\varepsilon\text{-eluder}}$ of $\mathcal{B}^d$ is the maximum length of a sequence $a'_1, \ldots, a'_s$ such that no element is $\varepsilon$-dependent on a prefix.

Consider the subset $S_a$ of $\{t \mid a \in \mathcal{A}^t, \mu_{X^t}(a) \neq a\}$ such that

$$\mathbf{1}\left[\max\left(C_{a,\mu_{X^t}(a)}\right) - \min\left(C_{a,\mu_{X^t}(a)}\right)\right) > \varepsilon].$$

Suppose for the sake of contradiction that

$$|S_a| > \left(\frac{4\beta_T}{\varepsilon^2} + 1\right) d_{\varepsilon\text{-eluder}}.$$

Then, there exists an element $t^*$ that is $\varepsilon$-dependent on $\frac{4\beta_T}{\varepsilon^2} + 1$ disjoint subsets of $S_a$: One can repeatedly remove sequences $a'_{\mu_{X^{t_1}}(a)}, \ldots, a'_{\mu_{X^{t_s}}(a)}$ of maximal length such that no element is $\varepsilon$-dependent on a prefix; note that $s \leq d_{\varepsilon\text{-eluder}}$ always. Let the subsets be $S_a^{(q)}$ for $q = 1, \ldots, \frac{4\beta_T}{\varepsilon^2} + 1$,

and let $\phi(a), \tilde{\phi}(a)$ be such that $\langle c_{\mu_{X^{t^*}}(a)}, \tilde{\phi}(a) - \phi(a)\rangle > \varepsilon$. The above implies that

$$\sum_{q=1}^{\frac{4\beta_T}{\varepsilon^2}+1} \sum_{t \in S_a^{(q)}} \langle c\mu_{X^t}(a), \tilde{\phi}(a) - \phi(a)\rangle^2 > 4\beta_T$$

by the definition of $\varepsilon$-dependence. But this is impossible, since the left-hand side is upper bounded by

$$\sum_{t=1}^T \langle c\mu_{X^t}(a), \tilde{\phi}(a) - \phi(a)\rangle^2 \le 4\beta_T$$

by the definition of the confidence sets. Hence it must hold that

$$|S_a| \le \left(\frac{4\beta_T}{\varepsilon^2} + 1\right) d_{\varepsilon\text{-eluder}}.$$

Now, it follows from the bound on the eluder dimension for linear bandits (Proposition 6 in [29]) that the bound of $\tilde{O}\left(\left(\frac{4\beta_T}{\varepsilon^2} + 1\right) d\log(1/\varepsilon).\right)$ holds. $\qquad\square$

**Lemma B.3.** *Let the confidence sets be defined as above (and in* MATCHLINUCB*). For any $a \in \mathcal{A}$, it holds that:*

$$\sum_{t|a\in\mathcal{A}^t, \mu_{X^t}(a)\neq a} \left(\max\left(C_{a,\mu_{X^t}(a)}\right) - \min\left(C_{a,\mu_{X^t}(a)}\right)\right) \le O(d(\log(T|\mathcal{A}|))\sqrt{T_a}),$$

*where $T_a$ is the number of times that agents is matched.*

*Proof.* Let's consider the set of confidence set sizes $\left(\max\left(C_{a,\mu_{X^t}(a)}\right) - \min\left(C_{a,\mu_{X^t}(a)}\right)\right)$ for $t$ such that $a \in \mathcal{A}^t, \mu_{X^t}$. Let's sort these confidence set sizes in decreasing order and label them $w_1 \ge \ldots \ge w_{T_a}$. Restating Lemma B.2, we see that

$$\sum_{t=1}^{T_a} w_t \mathbf{1}[w_t > \varepsilon] \le O\left(\left(\frac{4\beta_T}{\varepsilon^2} + 1\right) d\log(1/\varepsilon)\right). \tag{11}$$

for all $\varepsilon > 0$.

We see that:

$$\sum_{t|a\in\mathcal{A}^t, \mu_{X^t}(a)\neq a} \left(\max\left(C_{a,\mu_{X^t}(a)}\right) - \min\left(C_{a,\mu_{X^t}(a)}\right)\right) = \sum_{t=1}^{T_a} w_t$$

$$\le \sum_{t=1}^{T_a} w_t \mathbf{1}[w_t > 1/T_a^2] + \sum_{t=1}^{T_a} w_t \mathbf{1}[w_t \le 1/T_a^2]$$

$$\le \frac{1}{T_a} + \sum_{t=1}^{T_a} w_t \mathbf{1}[w_t > 1/T_a^2].$$

We claim that $w_i \le 2$ if $i \ge d\log(T_a)$ and $w_i \le \min(2, \frac{4\beta_T(d\log T_a)}{i - d\log T_a})$ if $i > d\log T_a$. The first part follows from the fact that we truncate the confidence sets to be within $[-1, 1]$. It thus suffices to show that $w_i \le \frac{4\beta_T(d\log T_a)}{i - d\log T_a}$ for $t \le d\log T$. If $w_i \ge \varepsilon > 1/T_a^2$, then we see that $\sum_{t=1}^{T_a} \mathbf{1}[w_t > \varepsilon] \ge i$, which means by (11) that $i \le O\left(\left(\frac{4\beta_T}{\varepsilon^2} + 1\right) d\log(1/\varepsilon)\right) \le O\left(\left(\frac{4\beta_T}{\varepsilon^2} + 1\right) d\log(T_a)\right)$ which means that $\varepsilon \le \frac{4\beta_T(d\log T_a)}{i - d\log T_a}$. This proves the desired statement.

Now, we can plug this into the above expression to obtain:

$$\sum_{t|a\in\mathcal{A}^t,\mu_{X^t}(a)\neq a}\Big(\max\big(C_{a,\mu_{X^t}(a)}\big)-\min\big(C_{a,\mu_{X^t}(a)}\big)\Big)$$

$$\leq \frac{1}{T_a}+\sum_{t=1}^{T_a}w_t\mathbf{1}[w_t>1/T_a^2]$$

$$\leq \frac{1}{T_a}+2d\log(T_a)+\sum_{i>d\log T_a}^{T_a}\min\left(2,\frac{4\beta_T(d\log T_a)}{i-d\log T_a}\right)$$

$$\leq \frac{1}{T_a}+2d\log(T_a)+2\sqrt{d\log T_a\beta_T}\int_{t=0}^{T_a}t^{-1/2}dt$$

$$=\frac{1}{T_a}+2d\log(T_a)+4\sqrt{dT_a\log T_a\beta_T}$$

We now use the fact that

$$\beta_T=O(d\log T+\frac{1}{T}\sqrt{\log(T^2|A|)}).$$

Plugging this into the above expression, we obtain the desired result. $\square$

With these facts, we are ready to prove Theorem 5.2.

*Proof of Theorem 5.2.* Like in the proof of Theorem 5.1, we divide into two cases, based on the event $E$ that all of the confidence sets contain their respective true utilities at every time step $t\leq T$. That is, $u_{c_1}(c_2)\in C_{c_1,c_2}$ for all $c_1,c_2\in\mathcal{C}$ at all $t$.

**Case 1: $E$ holds.** By Lemma 5.3, we know that the cumulative regret is upper bounded by

$$R_T\leq\sum_{t=1}^T\sum_{a\in\mathcal{A}^t}\Big(\max\big(C_{a,\mu_{X^t}(a)}\big)-\min\big(C_{a,\mu_{X^t}(a)}\big)\Big)$$

$$=\sum_{a\in\mathcal{A}}\sum_{t|a\in\mathcal{A}^t,\mu_{X^t}(a)\neq a}\Big(\max\big(C_{a,\mu_{X^t}(a)}\big)-\min\big(C_{a,\mu_{X^t}(a)}\big)\Big)$$

$$\leq\sum_{a\in\mathcal{A}}O(d\log(T|\mathcal{A}|)\sqrt{T_a}),$$

where the last inequality applies Lemma B.3 to the inner summand. We see that $\sum_{a\in\mathcal{A}}T_a=\sum_t|\mathcal{A}_t|\leq nT$ by definition, since at most $n$ agents show up at every round. Let's now observe that:

$$\sum_{a\in\mathcal{A}}\sqrt{T_a}\leq\sqrt{|\mathcal{A}|}\sqrt{\sum_{a\in\mathcal{A}}T_a}\leq\sqrt{|\mathcal{A}|nT},$$

as desired.

**Case 2: $E$ does not hold.** From Lemma B.1, it follows that:

$$\mathbb{P}[\phi(a)\in C_{\phi(a)}\quad\forall 1\leq t\leq T]\geq 1-1/(|\mathcal{A}|^3T^2).$$

Union bounding, we see that

$$\mathbb{P}[\phi(a)\in C_{\phi(a)}\quad\forall 1\leq t\leq T\forall a\in\mathcal{A}]\geq 1-1/(|\mathcal{A}|^2T^2).$$

By the definition of the confidence sets for the utilities, we see that:

$$\mathbb{P}[u(a,a')\in C_{a,a'}\quad\forall 1\leq t\leq T,\forall a,a'\in\mathcal{A}]\geq 1/(|\mathcal{A}|^2T^2). \tag{12}$$

Thus, the probability that event $E$ does not hold is at most $|\mathcal{A}|^{-2}T^{-2}$. In the case that $E$ fails to hold, our regret in any given round would be at most $4|\mathcal{A}|$ by the Lipschitz property in Proposition 4.4. Thus, the expected regret is at most $4|\mathcal{A}|^{-1}T^{-1}$ which is negligible compared to the regret bound from when $E$ does occur.

$\square$

## B.4 Proof of Lemma 5.4

**Lemma 5.4.** *Any algorithm that learns a stable matching with respect to unstructured preferences has an instance on which it has expected regret $\widetilde{\Omega}(|A|^{3/2}\sqrt{T})$ (as measured by Subset Instability).*

The idea behind proving this lemma is to show a lower bound for the easier problem of learning a maximum weight matching using utility difference as regret. By Proposition 4.4, this immediately implies a lower bound for learning a stable matching with regret measured by Subset Instability.

*Proof of Lemma 5.4.* Recall that, by Proposition 4.4, the problem of learning a maximum weight matching with respect to utility difference is no harder than that of learning a stable matching with respect to Subset Instability. In the remainder of our proof, we reduce a standard "hard instance" for stochastic multi-armed bandits to our setting of learning a maximum weight matching.

**Step 1: Constructing the hard instance for stochastic MAB.** Consider the following family of stochastic multi-armed bandits instances: for a fixed $K$, let $\mathcal{I}_\alpha$ for $\alpha \in \{1, \ldots, K\}$ denote the stochastic multi-armed bandits problem where all arms have 0-1 rewards, and the $k$-th arm has mean reward $\frac{1}{2} + \rho$ if $k = \alpha$ and $\frac{1}{2}$ otherwise, where $\rho > 0$ will be set later. A classical lower bound for stochastic multi-armed bandits is the following:

**Lemma B.4** ([4]). *The expected regret of any stochastic multi-armed bandit algorithm on an instance $\mathcal{I}_\alpha$ for $\alpha$ selected uniformly at random from $\{1, \ldots, K\}$ is $\Omega(\sqrt{KT})$.*

**Step 2: Constructing a (random) instance for the maximum weight matching problem.** We will reduce solving the above distribution over stochastic multi-armed bandits problems to a distribution over instances of learning a maximum weight matching. Let us now construct this random instance of the maximum weight matching problem. Let $|\mathcal{I}| = K$ and $|\mathcal{J}| = 10K \log(KT)$. Specifically, we sample inputs for learning a maximum weight matching as follows: For each man $i \in \mathcal{I}$, select $\alpha_i \in \{1, \ldots, K\}$ uniformly at random, and define $u_i(j)$ to be $\frac{1}{2} + \rho$ if $\lfloor (j-1)/\log K \rfloor = \alpha_i$ and $\frac{1}{2}$ otherwise. Furthermore, let $u_j(i) = 0$ for all $(i, j) \in \mathcal{I} \times \mathcal{J}$. Finally, suppose observations are always in $\{0, 1\}$ (but are unbiased).

The key property of the above setup that we will exploit for our reduction is the fact that, due to the imbalance in the market, the maximum weight matching for these utilities has with high probability each $i$ matched with some $j$ whom they value at $\frac{1}{2} + \rho$. Indeed, by a union bound, the probability that more than $10 \log(KT)$ different $i$ have the same $\alpha_i$ is at most

$$K \cdot \binom{K}{10\log(KT)} K^{-10\log(KT)} = O\big(K^{-4}T^{-4}\big).$$

Thus, with probability $1 - O(K^{-4}T^{-4})$, this event holds. The case where this event does not hold contributes negligibly to regret, so we do not consider it further.

**Step 3: Establishing the reduction.** Now, suppose for the sake of contradiction that some algorithm could solve our random instance of learning a maximum weight matching problem with expected regret $o(K^{3/2}\sqrt{T})$. We can obtain a stochastic multi-armed bandits that solves the instances in Lemma B.4 as follows: Choose a random $i^* \in \mathcal{I}$ and set $\alpha_{i^*} = \alpha$. Simulate the remaining $i$ by choosing $\alpha_i$ for all $i \neq i^*$ uniformly at random. Run the algorithm on this instance of learning a maximum weight matching, "forwarding" arm pulls to the true instance when matching $i^*$.

To analyze the regret of this algorithm when faced with the distribution from Lemma B.4, we first note that with high probability, all the agents $i \in \mathcal{I}$ can simultaneously be matched to a set of $j \in \mathcal{J}$ such that each $i$ is matched to some $j$ whom they value at $\frac{1}{2} + \rho$. Then, the regret of any matching is $\rho$ times the number of $i \in \mathcal{I}$ who are not matched to a $j$ whom they value at $\frac{1}{2} + \rho$. Thus, we can define the cumulative regret for an agent $i \in \mathcal{I}$ as $\rho$ times the number of rounds they were not matched to someone whom they value at $\frac{1}{2} + \rho$. For $i^*$, this regret is just the regret for the distribution from Lemma B.4. Since $i^*$ was chosen uniformly at random, their expected cumulative regret is at most

$$\frac{1}{K} \cdot o(K^{3/2}\sqrt{T}) = o(\sqrt{KT}),$$

in violation of Lemma B.4.

**Step 4: Concluding the lower bound.** This contradiction implies that no algorithm can hope to obtain $o(K^{3/2}\sqrt{T})$ expected regret on this distribution over instances of learning a maximum weight matching. Since there are $O(K\log(KT)) = \widetilde{O}(K)$ agents in the market total, the desired lower bound follows. $\square$