# OpenReview forum: "Learning Equilibria in Matching Markets from Bandit Feedback"
_NeurIPS.cc/2021/Conference — NeurIPS 2021 Spotlight_

### Official Review · Reviewer_SoA5 · 2021-07-06

**Rating:** 7
**Confidence:** 5

**Summary:**

The authors study stability in two-sided markets with transferable utilities where each agent has no a-priori knowledge of her own preference. They introduce the measure of 'Subsidy Instability' that is defined as the minimum total subsidy (extra payment to all the agents) that can stabilize a (matching, transfer) pair. They propose a UCB based primal-dual algorithm that tries to minimize the Subsidy Instability with bandit feedback for the above system. The proposed algorithm is proved to have a O(|A|^3/2\sqrt{T}) regret in a time horizon of T, where A is the set of all agents. They provide an orderwise matching  lower bound for this setting.

**Limitations And Societal Impact:**

The authors highlight that the paper lacks the study of fairness in utility allocation across agents of the proposed algorithm. They also discuss the scope of social impact of the paper.

**Main Review:**

Pros
- The paper studies an important problem for learning in matching markets, where agents are matched, and transfers are made to distribute the generated welfare in a stable manner.  This captures many important markets as mentioned in the paper.
- The introduction of the Subsidy Instability is interesting as it formulates the stability in two-sided markets in a centralized manner. The provided extensions show the flexibility of the proposed Subsidy Instability framework.
- The regret upper bound and the matching lower bound in the instance independent setting highlights the optimality (orderwise) of the proposed solution.
- The theoretical results in the extensions cover many important settings (e.g. instance dependent, linear reward), so the work is quite comprehensive within the scope of the paper.
- The writing is clear and easy to follow, and highlights the important aspect of the Subsidy Instability metric.

Drawbacks
-  How does the Subsidy Instability ensure (or fail to do so) the four points in the desiderata of [6]? I think parts of this is discussed in Appendix F, a summary of which is better suited for the main body of the paper.
- The global minimization of subsidy instability does not capture fairness, as acknowledged by the authors. Some discussion on how the Subsidy Instability per agent/max-min ratio  can grow under their proposed solution will be insightful.
- The (non)-uniqueness of the solution under Subsidy Instability needs more discussions. If the solutions are non-unique under UCB does the algorithm converge to a single solution? Some numerical validation may shed some light on this.
- Some discussion on the decentralization of the proposed Subsidy Instability will be useful as centralized algorithms are undesired due to privacy concerns.
- This paper does not mention some of the related works properly.
1. The difference from [6] which also study the matching with transfer setting should be elaborated. (see the first bullet)
2. The comparison with learning a stable matching in two sided market without transfer is incomplete. The results in the paper [5], [23], and [28] are not mentioned in the paper, although they are cited. Does the current formulation capture the heterogeneous ranking in both sides therein?
3. The application of a primal-dual algorithm with bandit feedback in prior works is not discussed. Please cite the appropriate works, e.g. 'An Asymptotically Optimal Primal-Dual Incremental Algorithm for Contextual Linear Bandits', Tirinzoni et al.
- The instance dependent regret does not capture the dependence on various reward gaps, but  just uses the minimum gap. Can this be made more finer (following Chen et al. and their gap definitions for super-arms) ? In general, the proofs of the extensions are somewhat incomplete (akin to sketches). I encourage the authors to complete the proofs in a technical report.

Minor
- In Lemma 4.4 it should be \mathcal{A} instead of A.


Note: All the citations are numbered according to the main paper.

**Time Spent Reviewing:**

10

---

> ### Author Response · Authors · 2021-08-10
> **Author Response**
>
> Thanks for the many helpful comments and questions! Below, we respond to the questions that you brought up.
>
> **Evaluation of Subsidy Instability along the axes proposed in [6].** Although the settings of [6] and our paper are different (see related work discussion below), we can still roughly evaluate Subsidy Instability along the axes proposed in [6]:
> 1. *Stability (with respect to transient preferences).* The matching with transfers chosen at each round is stable with respect to the UCB estimates of the preferences (referred to as “transient preferences” in [6]).
> 2. *Low regret.* We obtain near-optimal regret bounds.
> 3. *Fairness.* Our formulation of Subsidy Instability does not capture fairness. Finding an appropriate measure of per-agent “instability” would be an interesting direction for future research. (See also our discussion in Section 5.)
> 4. *Social welfare.* Our algorithm converges to a matching with optimal social welfare. Our regret bounds, measured by Subsidy Instability, also upper bound the gap to the optimal social welfare (see Proposition 3.3). In fact, we provide stronger social welfare guarantees than [6]—the algorithms in [6] only guarantee that welfare is within a 2-factor of optimal.
>
> We will add a discussion of these points to the paper.
>
> **Non-uniqueness of stable outcomes.** In the Shapley-Shubik model [30], stable outcomes are generally not unique. Even in the simple example—with Charlene, Percy, and Quinn—at the end of Section 2, there are multiple possible choices for the stable transfers. Which stable outcome we converge to depends on how the primal-dual pair $(X^*, p^*)$ in line 4 of ComputeMatch is chosen. That being said, our regret bounds hold regardless of this choice.
>
> **Decentralized setting.** Our definition of Subsidy Instability should provide a useful measure of distance from stability in this setting. This would be an interesting direction for future work.
>
> **Related work.** Thanks for the pointers to related work. We will add a more thorough discussion to the paper. Here is a summary of the main differences:
>
> *Comparison to [6].* While our paper and [6] study related questions, there are some crucial differences in the settings considered, which we highlight:
> - In [6], they focus on fixed, predetermined cost/transfer rules, such as “balanced transfers” and “pricing”. In contrast, we allow the platform to set arbitrary transfers between agents, which is closer to the characteristics of many online marketplaces.
> - The notion of stability in [6] does not consider agents negotiating arbitrary transfers: defecting agents must set their transfers according to a fixed, predetermined structure. In contrast, we follow the classical definition of stability for matching with transfers [30].
> - In [6], regret is based on utility difference, whereas we measure regret using Subsidy Instability.
>
> *Comparison to [5, 23, 28].* Our primary focus is the matching with transfers setting, though we do provide an extension of results to matching without transfers in Appendix F. We highlight several differences between our results in Appendix F and the settings considered by [5], [23], and [28].
> - These papers measure regret with respect to utility difference, while we focus on Subsidy Instability.
> - These works study a decentralized, competing bandits setting, where agents can compete over arms. We instead focus on matching coordinated by a centralized platform.
>
> In response to the reviewer’s question, our results in Appendix F apply to arbitrary heterogeneous rankings of agents, with unknown preferences on both sides.
>
> *Primal-dual bandits.* Thanks for the pointer to this literature; we will include a discussion of the connections in our paper.
>
>
>
>
> **Instance-dependent bounds.** Thanks for pointing out this interesting direction. We believe that we can sharpen our instance-dependent bounds following the argument of Chen et al., though we have not checked all of the details.
>
> **Proofs.** Thank you for reading our supplementary material. We will include more detailed proofs in the appendix of our paper.

---

> > ### Comment · Reviewer_SoA5 · 2021-08-11
> > **Response**
> >
> > Thanks for the detailed response. I find the response satisfactory. I decide to retain my score which was not influenced much by the drawbacks.
> >
> > Non-uniqueness of stable outcomes: By sticking to a specific primal-dual pair in line 4, will the solution (not just the reward) converge? Such convergence (or lack of it) may have implications in the decentralized setting. That is why I am curious.

---

### Official Review · Reviewer_noAP · 2021-07-16

**Rating:** 8
**Confidence:** 4

**Summary:**

The paper aims at learning stable matchings in settings with monetary transfers, when the learner does not initially know agents’ preferences and has to learn them over time. The authors provide a no-regret algorithm to do so, based on the idea of upper confidence bounds (UCB).

**Limitations And Societal Impact:**

Societal impact and fairness are not the primary concern of this paper. The authors acknowledge that computing a stable solution may come with unfairness, and that adding a fairness constraint is an interesting research direction.

**Main Review:**

Strengths:
- I like the subsidy instability notion. I think this is a good notion of instability as it measures exactly how far you are (monetarily) from a stable solution. Proposition 3.3 strengthens this notion, by showing that the instability notion is well-behaved and upper bounds the utility difference, which implies convergence to a socially optimal matching.
- The algorithms and results of the paper do not directly follow from applying UCB to the Lipschitz case. The authors use a primal-dual approach to compute the solutions they work with at each time step, in a way that they show reduces the size of the confidence interval at each step. The authors are then able to get an instance-independent regret of $\sqrt{T}$.
- The authors show a matching lower bound for the expected regret
- The authors also provide instance-dependent bounds.

Minor weakness:
- Overall, the algorithm still follows the general idea of UCB (even if there is some additional work to make it work in this setting) and does not seem super novel. Most of the novelty seems to come from understanding the properties of their instability measure/optimization programs.

Typos:
- Is there a sum missing in Equation 2 when defining the utility difference?

**Time Spent Reviewing:**

3-4

---

> ### Author Response · Authors · 2021-08-10
> **Author Response**
>
> Thanks for your comments! Good catch on the typo in equation (2). Indeed the definition of utility difference should read
> $$\max_{X’} \sum_{a\in\mathcal{A}} u_a(\mu_{X’}(a)) - \sum_{a\in\mathcal{A}} u_a(\mu_X(a)).$$

---

### Official Review · Reviewer_nPa8 · 2021-07-16

**Rating:** 9
**Confidence:** 5

**Summary:**

The paper considers two-sided platforms who seek to match users on
  both sides under uncertainty about user preferences. The focus is on
  the transferable utility setting, where the platform, in addition to
  orchestrating a matching, imposes transfers between each pair (as
  well as from the platform to the pair). After each matching, the
  platform receives noisy (bandit) feedback regarding the utilities
  received by each member in a matched pair. The goal of the platform
  is to learn a stable market outcome efficiently. To capture the
  (in)efficiency of learning, the authors introduce a measure of
  instability of any matching, namely subsidy instability, which is
  equal to the smallest total (non-negative) transfer the platform
  needs to make to the users so that matching is stable. The regret of
  any learning algorithm is then defined as the total subsidy
  instability accumulated over a time horizon.

  The authors first show that the subsidy instability is indeed an
  appropriate notion to capture learning efficiency, by providing
  natural economic interpretations as well as exhibiting a number of
  smoothness properties and bounds. The authors then propose a natural
  UCB-type algorithm, where each agent reports upper confidence bounds
  on their preferences and the platform computes the optimal stable
  matching and transfers under these reported preferences. The authors
  show that this algorithm achieves an expected regret that is of
  order $O(\sqrt{T\log T})$, and furthermore provide a matching lower
  bound. (The bounds also establish regret dependence on other model
  parameters.)

**Limitations And Societal Impact:**

 Yes.

**Main Review:**

* Pros:
  - The authors consider a natural learning setting that is
    practically important and relevant to many online platforms.
  - The main contribution of the work is in introducing/using the
    notion of subsidy instability to measure the efficiency of the
    learning algorithm. Since subsidy instability accurately captures
    the economic incentives of such platforms, exhibiting the
    performance of any algorithm in the terms of subsidy instability
    would be appealing to practitioners.
  - The resulting algorithm is simple and intuitive, and uses natural
    UCB-type ideas to achieve low regret. This simplicity would also
    be appealing in practice.
  - The paper is very well written, and the main results and the
    contributions are described with clarity.

Overall, the paper provides a clean and complete characterization of a practically relevant learning problem.

**Time Spent Reviewing:**

6 hours

---

> ### Author Response · Authors · 2021-08-10
> **Author Response**
>
> Thanks for your thoughtful comments and for your appreciation of our work!

---

### Official Review · Reviewer_PkCM · 2021-07-19

**Rating:** 7
**Confidence:** 3

**Summary:**

This paper proposes a generalization of the stable matching problem to the case of uncertain input on agent preferences, which are learned via bandit feedback, with a loss function aiming to minimize instability. To define this loss function, the notion of transferable utility is used, whereby monetary subsidies offered by the platform to agents, in addition to transfer between those agents, quantify the instability of a certain matching.

**Ethical Concerns:**

The paper has a strong similarity to another NeurIPS submission. (11554)

**Ethics Review Area:**

["Research Integrity Issues (e.g., plagiarism)"]

**Limitations And Societal Impact:**

The paper considers  a problem where the objective is to achieve social benefit.

**Main Review:**

The paper proposes novel algorithms that learn user preferences via optimism in the face of uncertainty from noisy user feedback in a multi-armed bandit setting, aiming to minimize a measure of instability in lieu of producing strictly stable matchings, providing near-optimal bounds. This economically motivated formulation extends to the case of structured preferences.

The work is well motivated by the case of data-driven markets, where agents may be incentivized to participate in a matching via monetary transfers. The concept of a stable matching, whereby the sum of agent utilities is maximized, is well generalized to the case of uncertainty, where users cannot report their full preferences, and instead those are learned via bandit feebdack.

**Needs Ethics Review:**

Yes

**Time Spent Reviewing:**

3

---

> ### Author Response · Authors · 2021-08-10
> **Author Response**
>
> Thanks for your thoughtful comments and for your appreciation of our work!

---

### Review · Ethics_Reviewer_okxf · 2021-08-09

**Recommendation:** This is in committee purview for two …

**Ethics Review:**

Reviewer PkCM flags this paper as being close to another submission. This is out of scope for ethical reveiw and I pass back to Committee to resolve this issue.

---

### Review · Ethics_Reviewer_6tND · 2021-08-10

**Recommendation:** Plagiarism concerned should be addres…

**Ethics Review:**

This paper was flagged for plagiarism concerns which I have not assessed as it is outside the scope of ethics review.

---

### Decision · Program_Chairs · 2021-09-27

**Decision:**

Accept (Spotlight)

**Comment:**

Reviewers unilaterally supported the paper - the approach is simple and the introduced subsidy idea aligns well with platform markets' realities, the characterization of the problem is reasonably complete, exposition is good, and the results are clean.  The most critical reviewer, SoA5, brings up solid points about regret and exposition/motivation around the subsidy, and in future versions of this work it would beneficial for the authors to address these concerns directly.